# Blood-brain barrier disruption defines the extracellular metabolome of live human high-grade gliomas

Cecile Riviere-Cazaux [1], Lucas P. Carlstrom[1], Karishma Rajani[1], Amanda Munoz-Casabella[1], Masum Rahman [1], Ali Gharibi-Loron[1], Desmond A. Brown[2], Kai J. Miller[1], Jaclyn J. White[3], Benjamin T. Himes[4], Ignacio Jusue-Torres[1], Samar Ikram[1], Seth C. Ransom[1], Renee Hirte[1], Ju-Hee Oh[5], William F. Elmquist[5], Jann N. Sarkaria [6], Rachael A. Vaubel [7], Moses Rodriguez[8], Arthur E. Warrington[1,8], Sani H. Kizilbash [9] & Terry C. Burns [1]✉

The extracellular microenvironment modulates glioma behaviour. It remains unknown if blood-brain barrier disruption merely reflects or functionally supports glioma aggressiveness. We utilised intra-operative microdialysis to sample the extracellular metabolome of radiographically diverse regions of gliomas and evaluated the global extracellular metabolome via ultra-performance liquid chromatography tandem mass spectrometry. Among 162 named metabolites, guanidinoacetate (GAA) was 126.32x higher in enhancing tumour than in adjacent brain. 48 additional metabolites were 2.05–10.18x more abundant in enhancing tumour than brain. With exception of GAA, and 2-hydroxyglutarate in IDH-mutant gliomas, differences between non-enhancing tumour and brain microdialysate were modest and less consistent. The enhancing, but not the non-enhancing glioma metabolome, was significantly enriched for plasma-associated metabolites largely comprising amino acids and carnitines. Our findings suggest that metabolite diffusion through a disrupted blood-brain barrier may largely define the enhancing extracellular glioma metabolome. Future studies will determine how the altered extracellular metabolome impacts glioma behaviour.

[1] Department of Neurological Surgery, Mayo Clinic, Rochester, MN, USA. [2] Neurosurgical Oncology Unit, Surgical Neurology Branch, National Institutes of Neurological Disorders and Stroke, National Institutes of Health, Bethesda, MD, USA. [3] Department of Neurological Surgery, Wake Forest Baptist Health, Winston-Salem, NC, USA. [4] Department of Neurological Surgery, Albert Einstein College of Medicine, Montefiore Medical Center, Bronx, NY, USA. [5] Brain Barriers Research Center, Department of Pharmaceutics, College of Pharmacy, University of Minnesota, Minneapolis, MN, USA. [6] Department of Radiation Oncology, Mayo Clinic, Rochester, MN, USA. [7] Department of Laboratory Medicine and Pathology, Mayo Clinic, Rochester, MN, USA. [8] Department of Neurology, Mayo Clinic, Rochester, MN, USA. [9] Department of Oncology, Mayo Clinic, Rochester, MN, USA. ✉email: burns.terry@mayo.edu

Astrocytomas and oligodendrogliomas comprise most adult primary malignant brain tumours and remain incurable, regardless of the best available therapies[1]. Isocitrate dehydrogenase (IDH) mutations occur in all oligodendrogliomas and a subset of astrocytomas, affording improved prognosis due to typically slower growth and more favorable response to standard-of-care radiation and alkylating chemotherapy[2]. Glioma heterogeneity hampers therapeutic generalisations across diverse patient cohorts. Moreover, diverse genetic phenotypes accumulate within individual patients' tumour ecosystems, hampering therapeutic efforts to target glioma-associated mutations[3,4]. Conversely, molecularly diverse gliomas may resort to similar metabolic pathways in the setting of nutrient deprivation, hypoxia, and genotoxic stress[5,6]. Although glioma metabolism is increasingly scrutinised for therapeutic targets, few strategies currently exist to interrogate the metabolic microenvironment of human gliomas in situ, and there is a relative paucity of global metabolomic data from live human gliomas.

Microdialysis has been used to quantify human extracellular biomarkers of traumatic and hypoxic brain injury in neurocritical care units. This is most commonly performed in the post-operative setting with low-molecular weight catheters, including an FDA-approved system consisting of 20 kDA catheters perfused at 0.3 μL/min using the M-dialysis 106 pump. Microdialysis is also a well-established method to quantify central nervous system (CNS) drug delivery in early phase clinical trials[7–9] and has been used longitudinally to study metabolites present in human gliomas when compared to adjacent brain[10,11]. While a limited subset of patients may be willing to undergo post-operative microdialysis, it is usually hoped that there will be minimal tumour left to sample following surgery, often preventing sampling from multiple regions of the glioma. As such, intra-operative microdialysis may be of interest for evaluating diverse regions of gliomas. However, while some studies have utilised intraoperative low molecular weight microdialysis (8–20 kDA) to quantify select extracellular metabolites[12–14], no study thus far has deployed intra-operative microdialysis to characterise the global extracellular glioma metabolome, nor to compare the metabolome of contrast-enhancing versus non-enhancing tumour regions. We used variable flow-rate pumps set to 2 μL/min and high molecular weight catheters (100 kDa) to maximise the volume of microdialysate recovered during 15 surgeries in 14 patients. We report that contrast-enhancing (radiographically blood-brain barrier (BBB) disrupted) regions of high-grade gliomas (HGGs) exhibited a conserved extracellular metabolome across both IDH-mutant and IDH-WT astrocytomas, similar to findings from other groups utilising intra-or-post-operative microdialysis[13,15,16]. However, comparison to the radiographically BBB intact (non-enhancing) portions of these tumours revealed that this enhancing glioma signature was significantly enriched for plasma-associated metabolites, suggesting that BBB disruption may contribute to the glioma extracellular metabolome.

## Results

### Elevated D-2-Hydroxyglutarate (D-2-HG) in IDH-mutant tumour microdialysate.
Intra-operative microdialysis can enable sampling of diverse regions within the live extracellular glioma microenvironment during standard-of-care glioma resections. However, low molecular weight catheters preclude collection of most extracellular analytes; slow flowrates combined with short intra-operative sampling times prevent acquisition of enough microdialysate volumes to enable multi-omic analyses. We hypothesised that we could safely and feasibly deploy intra-operative high molecular weight (HMW, 100 kDa) microdialysis

using an elevated perfusion rate of 2 μL/min[14] to recover a usable volume of microdialysate containing diverse analytes, including metabolites, during a standard-of-care surgery. To that end, we initiated our intra-operative microdialysis trial in an initial cohort of five patients. To evaluate the feasibility of the intra-operative microdialysis approach, we aimed to measure microdialysate levels of the well-characterised and dialysable oncometabolite, D-2-hydroxyglutarate (D-2-HG), known to be produced by IDH-mutant gliomas[17]. As such, patients were enrolled whose tumours could plausibly be IDH-mutant. Subsequent pathology revealed that three of these initial five patients indeed had IDH-mutant gliomas (Oligo[2], Oligo[3]1, and Astro[4-mut]1); two patients were found to have IDH-wild type (WT) glioblastomas (GBM[WT]1, GBM[WT]2) (Table 1). To determine if D-2-HG levels in intra-operatively acquired microdialysate reflected IDH status, we utilised 10 μL of a 40 μL microdialysate sample from each catheter to quantify D-2-HG and L-2-HG via targeted metabolomic analysis. D-2-HG was elevated in microdialysate from IDH-mutant as compared to IDH-wild-Type (WT) tumours (median: 27.29 versus 0.63 μM) (Supplementary Fig. S2A), levels of L-2-HG were not different between groups (Supplementary Fig. S2B). Patient Astro[4-mut]1 yielded a > 1000x difference in D-2-HG between non-enhancing tumour (FLAIR) and brain adjacent to tumour (831.80 versus 0.68 μM). These data demonstrated the feasibility of intraoperative HMW microdialysis to sample a dialysable oncometabolite during standard-of-care surgery for tumour resection.

Since IDH status has been reported to impact the metabolism of IDH-mutant gliomas, we further analysed these samples via untargeted metabolomic analyses--an increasingly leveraged strategy to gain more comprehensive insights regarding the global metabolome. We utilised 20 μL of microdialysate from each catheter for untargeted metabolomic analysis via the Metabolon platform[18]. Among hundreds of other metabolites, untargeted analysis reported total 2-HG. When plotted against the combined stereoisomer (D-2-HG + L-2-HG) levels from targeted analyses, a robust correlation ($R^2 = 0.9989$) between the two platforms was observed across a 2500-fold concentration range (Supplementary Fig. S2C). Although limited sample volume precluded repeating this approach for other metabolites, these data supported the potential for untargeted metabolomic analyses to generate at least relatively quantifiable data.

Analysis of 2-HG peak area in a second cohort comprising 5 IDH-mutant and 5 IDH-WT cases, including one patient's repeat surgery, confirmed increased levels in IDH-mutant tumours when compared to IDH-WT tumours (21.41x average normalised peak area in IDH-mutant tumour catheters compared to IDH-WT tumour catheters) (Fig. 1c). However, one patient with an unusual right frontal H3K27M-mutant infiltrative grade 4 astrocytoma also demonstrated moderately elevated 2-HG levels despite Next Generation Sequencing-confirmed IDH-WT status. This one apparent "outlier" illustrates that elevated extracellular 2-HG is not infallibly specific for IDH-mutant gliomas and is consistent with prior reports of occasionally elevated 2-HG in IDH-WT gliomas[19].

Additionally, in two patients with grade 4 IDH-mutant astrocytomas, the extracellular abundance of 2-HG was several-fold less in enhancing than non-enhancing tumour (Fig. 1d, FC (E/NE): 0.017x and 0.20x for patients Astro[4-mut]1 and Astro[4-mut]2, respectively). This observation raised the possibility that 2-HG could be lost down its concentration gradient into systemic circulation. Indeed, elevated 2-HG was previously reported in venous outflow of IDH-mutant high-grade gliomas[20]. An alternate explanation could be that enhancing tumour regions from IDH-mutant gliomas generate less 2-HG due to changes in the IDH metabolism of the tumour core. Arguing against this

**Table 1. Intraoperative microdialysis patient characteristics.**

| Patient ID | Pathology | IDH Status | Other molecular/genetic features | Catheter X | Catheter Y | Catheter Z |
|---|---|---|---|---|---|---|
| Oligo[2] | Oligodendroglioma WHO 2 | Mutant | 1p/19q co-deletion; *TERT*-mutant | NE | NE | NE |
| Oligo[3]1 | Oligodendroglioma WHO 3 | Mutant | 1p/19q co-deletion; *TERT*-mutant | NE | NE | B |
| Oligo[3]2 | Oligodendroglioma WHO 3 | Mutant | 1p/19q co-deletion; *TERT*-mutant | NE | NE | B |
| Oligo[3]3 | Oligodendroglioma WHO 3 | Mutant | 1p/19q co-deletion; *TERT*-mutant | NE | NE | B |
| GemAstro[3]-mut | Gemistocytic Astrocytoma WHO 3 | Mutant | None remarkable | NE | NE | B |
| Astro[4]-mut] | Astrocytoma WHO 4 (Recurrent) | Mutant | *MGMT* non-methylated; loss of *CDKN2A/B*; complex karyotype | E | NE | B |
| Astro[4]-mut2 | Astrocytoma WHO 4 | Mutant | *MGMT* non-methylated; loss of *CDKN2A/B* | E | NE | B |
| Astro[4]-mut3 | Astrocytoma WHO 4 (Recurrent) | Mutant | Indeterminate *MGMT* methylation | E | E[b] | B |
| H3K27M-Astro[4]-WT | Infiltrating Astrocytoma WHO 4 | Wild-type | *H3K27M, TERT*-mutant; *MGMT* non-methylated | E | NE | B |
| GBM[WT]1 | Glioblastoma | Wild-type | *MGMT* non-methylated; loss of *CDKN2A/B; TERT*-mutant | E | NE | B |
| GBM[WT]2 | Glioblastoma (Primary) | Wild-type | Indeterminate *MGMT* methylation[a] | E | NE | B |
| GBM[WT]2B | Glioblastoma (Recurrent) | Wild-type | Indeterminate *MGMT* methylation[a] | E |  | B |
| GBM[WT]3 | Glioblastoma | Wild-type | *MGMT* non-methylated | E | NE | B |
| GBM[WT]4 | Glioblastoma (Recurrent) | Wild-type | *MGMT* non-methylated; *EGFR* amplification; loss of *CDKN2A/B* | E | NE | B |
| MolecGBM[WT] | Molecular Glioblastoma | Wild-type | Gain of ch. 7, loss of ch. 10; *TERT*-mutant; *MGMT* non-methylated | NE | NE | B |

E: enhancing tumor; NE: non-enhancing tumor; B: brain adjacent to tumor.
The patient IDs and final pathologic diagnoses, including IDH- status and other available key molecular features, are summarized together with the catheter locations. Additional specifics regarding each patient, tumour, and catheter are available in Supplementary Table 1.
[a]Diagnosis of *IDH*-WT GBM was reached for patient GBM[WT]2/2B without need for further molecular studies.
[b]Region of minimal enhancement, suggestive of tumour necrosis.

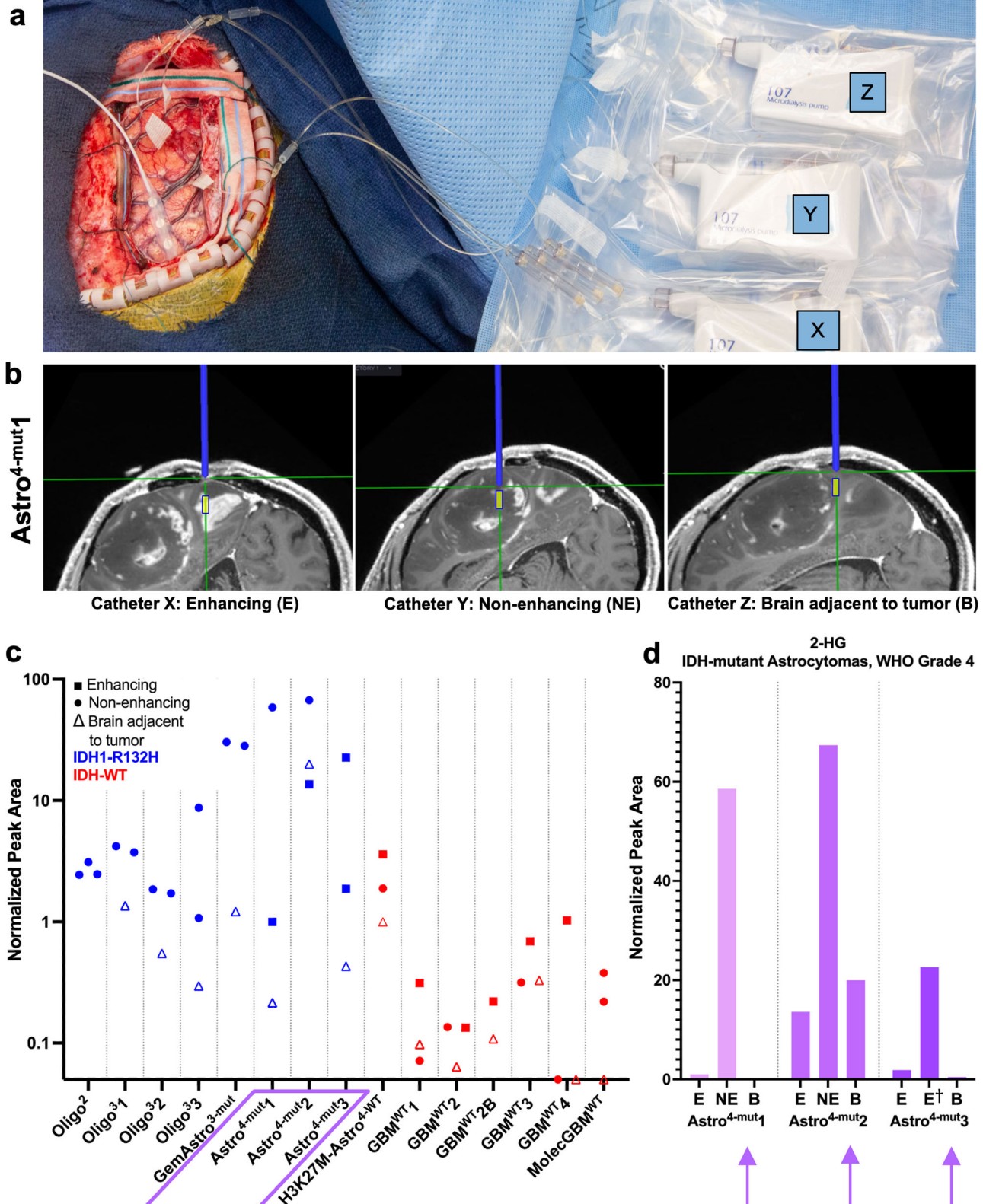

**Fig. 1 Intraoperative microdialysis set-up and D-2-hydroxyglutarate (D-2-HG) detection. a** A representative intraoperative photograph (Oligo[2]) demonstrating the experimental setup, including placement of three 100 kDA microdialysis catheters, as well as pumps (X, Y, Z) and collection vials within the surgical field. **b** Illustrative intraoperative trajectory views captured from the Neuronavigation system (Astro[4-mut]1) demonstrate the planned trajectory (blue line) toward the intended target location for each 10 mm microdialysis membrane (indicated by yellow box) in enhancing (E, Catheter X) and non-enhancing tumour (NE, Catheter Y), in addition to relatively normal brain adjacent to tumour (B, Catheter Z). **c** 2-HG peak areas from UPLC-MS/MS were measured in the microdialysate from each of the 44 catheters. IDH status is indicated as mutant (blue) or wild type (red). Symbol shape indicates catheter placement location (see legend). **d** The normalized 2-HG peak areas from three patients with enhancing grade 4 *IDH*-mutant astrocytomas are shown on a linear scale.

alternate explanation, a third patient with an enhancing grade 4 *IDH*-mutant astrocytoma showed enhancement throughout the lesion, precluding placement of a catheter in non-enhancing tumour. Although both tumour catheters were placed within the margins of contrast-enhancing tumour, the second tumour catheter targeted a region of less intense enhancement, suggestive of poorer central perfusion or relative necrosis (E†). Despite its placement in presumably poorly perfused tumour core, this catheter recovered 12.1-fold more 2-HG than the catheter in the most avidly contrast-enhancing region, suggesting that *IDH*-mutant metabolism was still present in the tumour core. Collectively, data from these three patients suggested that 2-HG may be lost from the extracellular compartment in regions of contrast-enhancing tumour due to increased diffusion across a disrupted blood-brain barrier (Fig. 1d).

**Clustering of catheters: correlation analysis of the extracellular global metabolome.** Having established that intra-operative HMW microdialysis could detect the relative abundance of an established oncometabolite of *IDH*-mutant gliomas, we next sought to evaluate the global extracellular metabolome across our patients' 44 catheters. To ensure comparison and reproducibility of metabolite levels across patients, we analysed 162 named metabolites that were present in at least 90% of the catheters ($\geq 40/44$ catheters). We first asked how patient's catheters correlated to each other to identify possible patterns in the global metabolome based on patient identity, radiographic location, *IDH* mutation, and primary vs. recurrent status. Spearman correlation of all 44 catheters revealed that all samples from each patient with an oligodendroglioma ($n = 3$ patients, 9/9 catheters) clustered based on patient identity. Conversely, 9/10 catheters from enhancing regions of WHO grade 4 astrocytomas clustered together (Fig. 2a). Clustering of non-enhancing tumour regions was inconsistent, with some catheters clustering near the patient's own enhancing catheters (patients Astro[4-mut]1, Astro[4-mut]2, and Astro[4-mut]3), while others clustered with brain catheters (Fig. 2a), particularly those from a completely non-enhancing tumour (MolecGBM[WT]1). Although Patient GemAstro[3-mut]'s tumour did have enhancing components, these areas were extremely fibrous, hampering catheter placement. As such, both catheters were placed into non-enhancing regions. Interestingly, however, this patient's catheters clustered with the enhancing tumour catheters, potentially suggesting flow of metabolites from nearby enhancing tumour regions (Fig. 2a). Contrary to our expectations based on recently published tissue metabolomic studies, no obvious clustering was observed based on *IDH* status[21], nor based on recurrence. Indeed, one patient's enhancing catheter from their primary resection (GBM[WT]2: Catheter X) was most strongly correlated with the enhancing catheter from their repeat resection (GBM[WT]2B: Catheter X). In sum, results of correlation analyses suggest that the extracellular metabolome of molecularly diverse contrast-enhancing astrocytomas clustered by catheter location, whereas that of oligodendrogliomas clustered by patient. Perhaps unsurprisingly, the extracellular metabolome of non-enhancing tumour regions of WHO grade 4 astrocytomas appeared to exist on a spectrum, appearing in some cases more similar to the enhancing tumour and in other patients, closer to that of the "brain" catheter. However, visual inspection of relative metabolite abundance as a function of catheter demonstrated substantial heterogeneity across patients within each region (Fig. 2b).

**A reproducible metabolome of enhancing glioma versus brain.** Given the substantial metabolic heterogeneity between patients, we asked if each patient could serve as their own control to identify the extracellular tumour metabolome. As the Spearman correlation had

suggested clustering of enhancing tumour catheters (Fig. 2a), we began by comparing catheters in enhancing tumour as compared to adjacent brain. For each patient, metabolites were ranked from highest to lowest fold change based on metabolite peak area in enhancing tumour (E) versus brain adjacent to tumour (B). Although the magnitude of these fold changes varied between patients (Supplementary Fig. S3), the rank of enhancing tumour versus brain-associated metabolites appeared similar in 3/3 patients with enhancing tumours from our initial cohort (Supplementary Fig. S4A, first and second cohort). We used the average ranks from these 3 patients to then evaluate the reproducibility of findings in the second cohort of 5 new patients and one patient's recurrent (GBM[WT]2B) disease 10 months later. The ranked order of metabolites in the enhancing versus brain comparison appeared visually similar for all patients in both cohorts when ranked based on the average of the 9 cases (Fig. 3ai, ii). Additionally, paired enhancing versus non-enhancing tumour catheters were available from 7 patients. Similar rank-based analyses were performed for enhancing versus non-enhancing tumour which revealed a surprisingly conserved rank of enhancing-to-non-enhancing metabolites across patients (Fig. 3bi, ii; results presented as first and second cohort in Supplementary Fig. S4B). Conversely, the distribution of non-enhancing catheter-associated metabolites appeared less consistent (Fig. 3ci; results presented as first and second cohort in Supplementary Fig. S4C). Despite this diversity, when this non-enhancing-to-tumour metabolite distribution was queried in five patients from whom only non-enhancing tumour and brains catheters were available, a similar ranked metabolome was observed in two patients, Oligo[3]1 and GemAstro[3-mut] (Supplementary Fig. S5). These data demonstrated that metabolic diversity exists among patients with oligodendrogliomas and suggests that there could be recurrent metabolic patterns in some, but not all, non-enhancing tumours.

To more rigorously and quantitatively compare metabolic phenotypes between catheters and between patients, we utilised enrichment analysis (EA) to statistically identify similarities between and across patients' catheters. Although more commonly applied to gene-based analyses, we repurposed this for metabolomic analysis. Using EA, we selected the top and bottom 35 metabolites based on fold change from each patient's tumour vs. brain or enhancing vs. non-enhancing tumour ranked metabolite list ("metabolite sets") (Supplementary Fig. S1). We then determined where these metabolites fell in the ranked metabolite list of tumour versus brain or enhancing vs. non-enhancing tumour for the remaining patients. "Positive enrichment" meant that the metabolites fell at the tumour-end of the ranked list, while "negative enrichment" indicated that they fell at the brain-end of the ranked list. In patients with enhancing astrocytomas, each patient's individual enhancing tumour signature could be robustly and positively identified in the enhancing tumour of the other patients, suggesting a convergent glioma metabolome (Fig. 3aii). Conversely, each patient's brain signature could be found at the "brain" end of every other patients' ranked metabolite list (Fig. 3aii). Leveraging these individual patient-level data, these analyses proved highly robust across patients with a False Discovery Rate (FDR) of $<1 \times 10^{-5}$ for most interpatient comparisons of enhancing tumour versus brain metabolites (Fig. 3aii, Supplementary Table 2). For independent reference, the previously published post-operative tumour versus adjacent brain microdialysate metabolome from Björkblom et al.[16] was evaluated and found to be present in the enhancing catheters of astrocytomas (FDR of 0.002 or less, Supplementary Table 2). Indeed, among 24 reported brain- or tumour-associated metabolites, all but one metabolite revealed a reproducible pattern in our data set (Supplementary Fig. S6). These data confirmed the robust nature of the extracellular enhancing glioma metabolome, being readily and comparably discernible in both the intra-operative and post-operative settings.

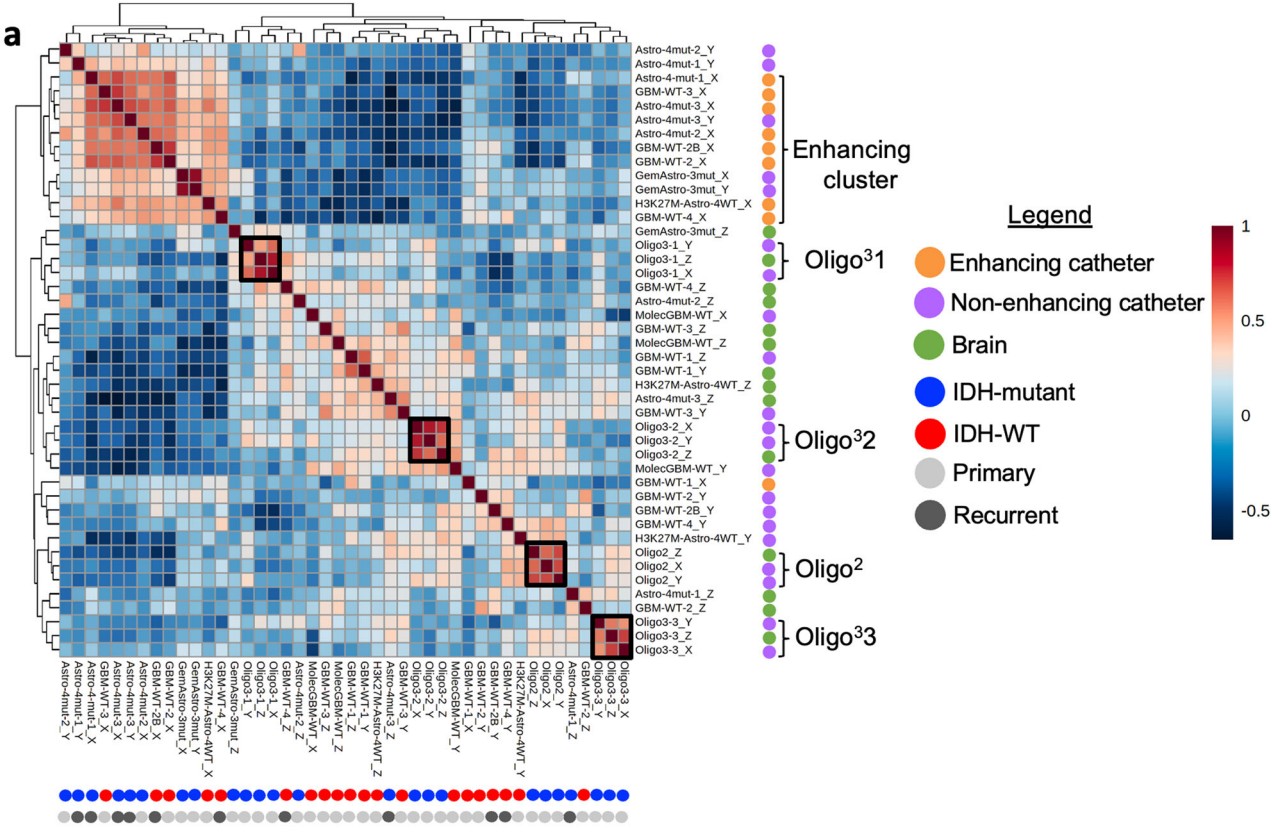

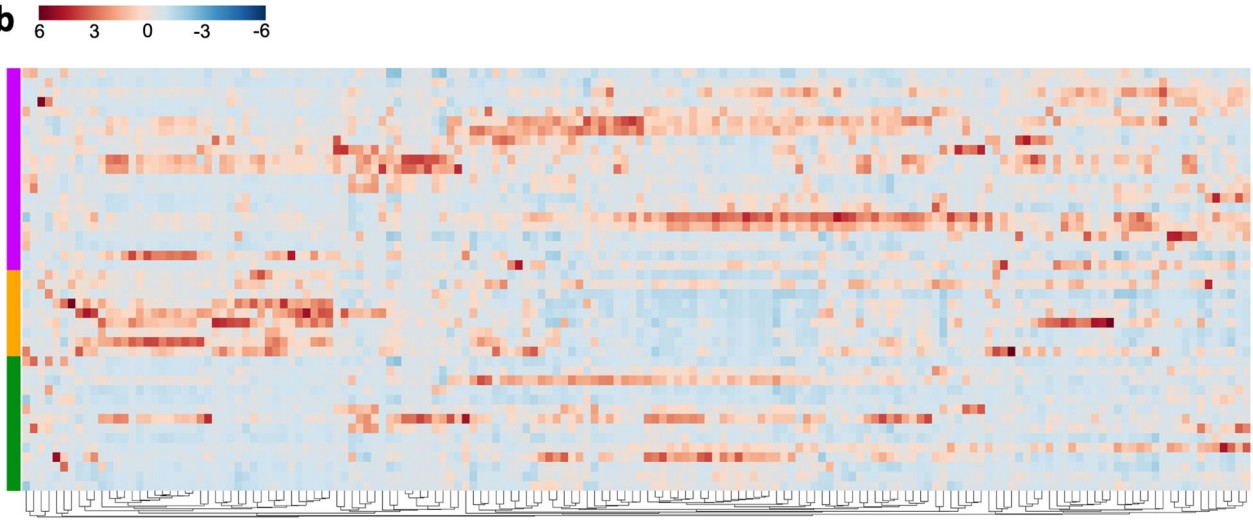

**Fig. 2 Microdialysate samples cluster based on patient identity and catheter location. a** Spearman correlation map of the 44 catheters from all 15 microdialysis cases (14 patients). Dot colours indicate catheter location, IDH status, and primary or recurrent status. (Minimal correlation: −0.7, maximal correlation: 1). **b** Hierarchical clustering heat map (Ward method, Euclidean distance measure) of the 44 catheters based on 162 metabolites present in at least 90% (40/44) catheters. (Normalised abundance by MetaboAnalyst: minimum, −6 to maximum, 6).

Enrichment analyses of enhancing versus non-enhancing catheters again demonstrated that enhancing catheters were metabolically separate from their non-enhancing catheter counterparts and that this could be consistently found across most patients (most FDRs $<1 \times 10^{-5}$, except GBM$^{WT}$1, Fig. 3bii; Supplementary Table 3). As expected from Fig. 3ci, non-enhancing catheters were rarely enriched for each other when compared to brain (Fig. 3cii, Supplementary Table 4). Overall,

these data suggest that the extracellular metabolome of enhancing glioma is distinct from and more consistent than that of non-enhancing glioma.

**Metabolites of the enhancing and non-enhancing glioma extracellular microenvironments.** After demonstrating that enhancing astrocytomas exhibit a conserved extracellular metabolome via rank-based and formal enrichment analyses (Fig. 4a),

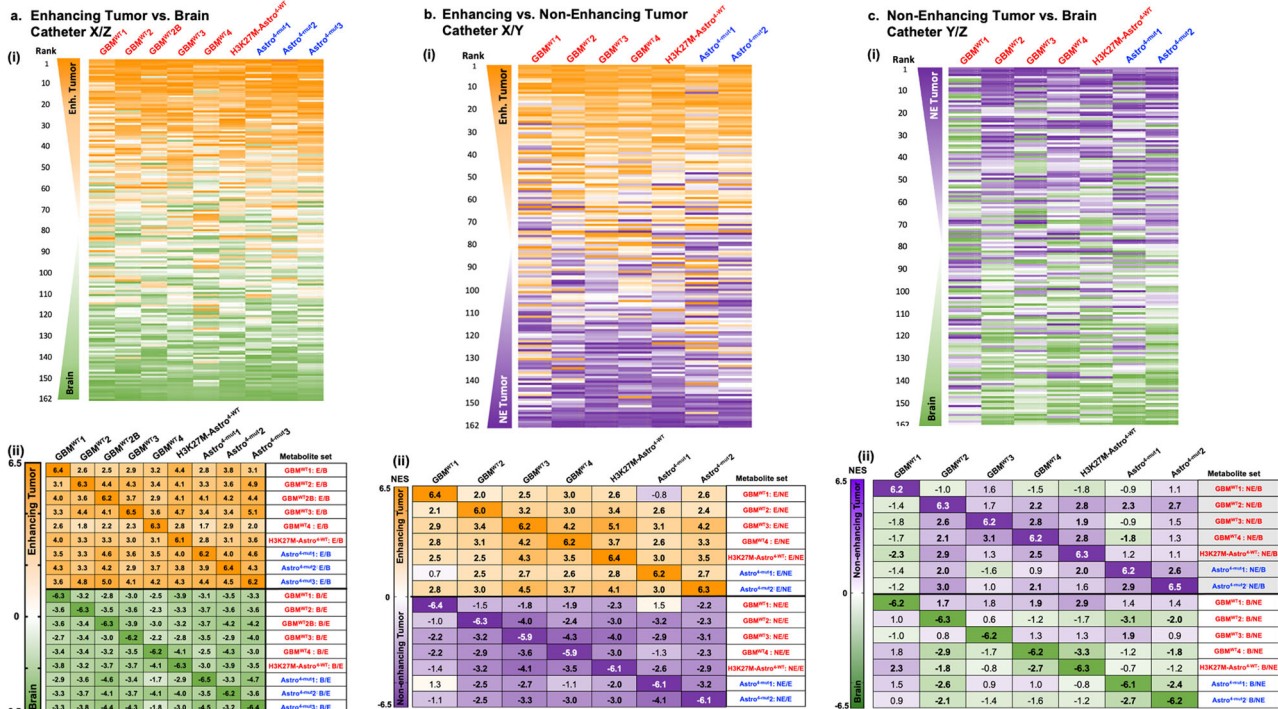

**Fig. 3 Metabolic signatures of enhancing versus non-enhancing tumours versus brain. a** (i) 162 metabolites present in at least 40/44 catheters were ranked according to the enhancing tumour-versus-brain fold change in each patient. The rank order of each metabolite in each 2-catheter tumour/brain comparison (e.g., catheter X versus Z) is conveyed as a heat map from 1 (orange, enhancing tumour) to 162 (green, brain). Metabolites are listed based on the average of ranks of enhancing vs. brain across nine cases. (ii) Normalised enrichment scores (NES) are shown based upon enrichment analysis for the top 35 enhancing tumour (E) and brain (B)-associated metabolites in the ranked metabolite of enhancing tumour vs. brain (Cath. X/Z) (bold: significant at FDR < 0.05; Complete FDR details available in Supplementary Table 2). See Supplementary Fig. 1 for graphical depiction of how enrichment analyses are performed by repurposing the Gene Set Enrichment Analysis software. **b** Using the same method as in 3 A(i), the rank order of each metabolite in enhancing-versus-non-enhancing catheter metabolite lists (Catheter X and Y) is depicted as a heat map in the 7 patients for whom both enhancing and non-enhancing catheters were present. Metabolites are listed based on the average of ranks of enhancing versus non-enhancing tumour across seven patients. (ii) As in 3 A(ii), using enhancing versus non-enhancing tumour metabolite ranked lists and metabolite sets. NES are shown for the enrichment of each E vs. NE ranked list for other E vs. NE metabolite sets (bold: FDR < 0.05; Complete FDR details available in Supplementary Table 3). **c** (i) As in 3A-B, the ranked order of metabolites in non-enhancing versus brain metabolite lists is shown as a heat map. Metabolites are ranked based on the average rank of NE tumour to brain across 7 patients. (ii) As in prior enrichment analyses, with non-enhancing versus brain ranked lists and metabolite sets and their respective NESs across patients (bold: FDR < 0.05; Complete FDR details available in Supplementary Table 4).

we next asked which metabolites define this extracellular microenvironment across the full cohort of all 9 surgeries. The Wilcoxon rank sum test for non-parametric distributions, paired enhancing vs. brain catheters revealed 48 significantly elevated metabolites in the enhancing tumour catheter and 22 significantly elevated metabolites in the brain catheter using cutoff criteria of $p$-value ≤ 0.05 and FC ≥ 2 (Fig. 4a; Table 2). The top enhancing astrocytoma metabolite was guanidinoacetate (GAA) (E/B = 126.32x), which, to our knowledge, has not previously been reported in the context of glioma. The enhancing glioma signature was also defined by greater relative abundance of proline (FC = 10.18x), carnitine-family metabolites (deoxycarnitine, carnitine, and acetylcarnitine at FC = 8.71x, 7.65x, and 7.47x, respectively), and N1-methyl-2-pyridone-5-carboxamide (2-PY) (FC = 7.18x) (all at $p < 0.005$). Metabolites defining the brain metabolome included arabitol (FC = 0.04) and arabonate (FC = 0.08), in addition to canonical brain-associated metabolites, such as N-acetylaspartate (NAA)[22] (all $p < 0.005$).

We then asked if these metabolites were unique to enhancing tumour when compared to non-enhancing tumour for each patient (Fig. 4b). Again using a Wilcoxon rank sum test for non-parametric distributions, twenty-eight metabolites were significantly higher in the enhancing glioma metabolome, while twenty metabolites were greater in the non-enhancing glioma

metabolome (Table 2). Interestingly, 27/28 enhancing glioma metabolites had also been found in the enhancing glioma vs. brain comparison, again demonstrating a robust extracellular metabolome of enhancing glioma (relative abundance of representative metabolites shown in Fig. 4c). Of the twenty non-enhancing versus enhancing glioma metabolites, 15 overlapped with brain (versus enhancing tumour). These data suggest the continued presence of brain-associated metabolites within non-enhancing portions of glioma. Indeed, when non-enhancing glioma catheters were compared to brain (Fig. 4d), no metabolites were unique to the brain metabolome, while fifteen metabolites were higher in non-enhancing tumour, including GAA (FC NE/B = 13.63x). Overall, these data demonstrate that a specific set of metabolites define the extracellular microenvironment of enhancing HGGs when compared to non-enhancing glioma and brain adjacent to tumour.

**Plasma in the glioma extracellular metabolome.** By definition, contrast-enhancing glioma regions have a more disrupted blood-brain barrier than non-enhancing regions[23], facilitating solute diffusion between the intravascular compartment and tumour microenvironment. Our data demonstrated a similar metabolic signature of enhancing, but not non-enhancing portions, of gliomas when compared to brain (Fig. 3a, c), and metabolic

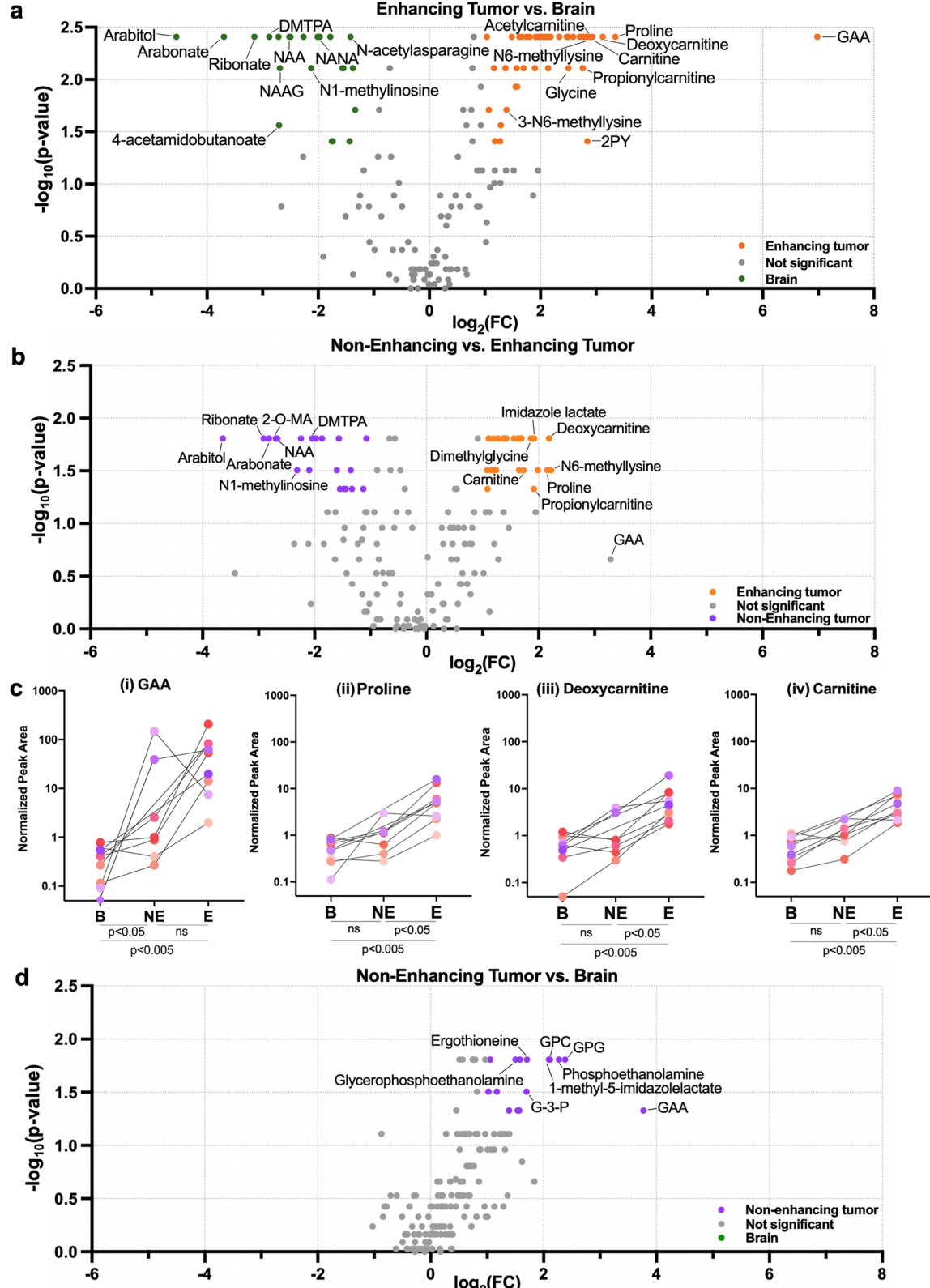

differences between enhancing vs. non-enhancing portions of gliomas (Figs. 3b, 4b). As such, we hypothesised that enhancing glioma metabolites may originate from plasma preferentially entering regions of greater BBB disruption where there is decreased resistance to diffusion. To test this hypothesis, we utilised a ranked metabolite list of "bloodiness" from an identically paired clean versus bloody CSF sample (see methods). In the 7 patients for whom paired enhancing and non-enhancing microdialysates were obtained, we first compared the relative enrichment of enhancing versus non-enhancing brain catheters for bloody versus clean CSF using each patient's ranked enhancing versus non-enhancing list (Cath. X/Y). Enrichment analyses

**Fig. 4 Differentially abundant extracellular metabolites in the enhancing glioma, non-enhancing glioma, and adjacent brain comparisons.** Wilcoxon signed-rank tests and fold-changes were calculated between (**a**) enhancing tumour vs. brain ($n = 9$ paired catheters), (**b**) enhancing versus non-enhancing tumour ($n = 7$ paired catheters), and (**d**) non-enhancing tumour versus brain ($n = 7$ paired catheters) were utilised to construct a volcano plot (cut-offs for significance: p-value $\leq 0.05$; FC $\geq 2$). The normalised peak areas for the top four most differentially abundant extracellular metabolites in enhancing tumour vs. brain are shown in (**c**) for the 9 cases, including the peak area in non-enhancing tumour for the seven patients for whom this catheter was available. See Table 2 for significant metabolites in each comparison (E/B, E/NE, and NE/B). 2-O-MA = 2-O-methylascorbic acid, DMTPA = 2,3-dihydroxy-5-methylthio-4-pentenoic acid, G-3-P=glycerol-3-phosphate, GAA guanidinoacetate; GPC glycerophosphocholine, GPG glycerophosphorylglycerol, NAA n-acetylaspartate; NAAG n-acetylaspartylglutamate, NANA n-acetylneuraminate.

demonstrated significant enrichment in 7/7 patients for at least one of (i) enhancing tumour enriched in bloody CSF, or (ii) non-enhancing tumour enriched in clean CSF; 5/7 patients showed significant enrichment in both analyses (FDR < 0.05) (Fig. 5a). Evaluation of specific enhancing versus non-enhancing tumour-associated metabolites from Fig. 4b revealed higher abundance of bloody CSF-associated metabolites in enhancing tumour (Fig. 5b). As expected, performing enrichment analyses on enhancing glioma versus brain ranked list again demonstrated that enhancing glioma catheters were enriched for bloody CSF in 9/9 patients (Supplementary Fig. S7). Results in non-enhancing tumour suggested a modest enrichment of bloody CSF-associated metabolites in some catheters, perhaps suggestive of either some sub-radiographic BBB disruption[23] or metabolite diffusion from adjacent enhancing regions.

As it is technically challenging to obtain intrapatient paired bloody versus clean CSF samples, we evaluated the reproducibility of these results by performing the same analyses using pooled, rather than paired, bloody versus clean CSF samples ($n = 7$ each; see methods). The pooled bloody versus clean CSF ranked list was positively enriched for bloody metabolites in the paired CSF sample and negatively enriched for clean metabolites (FDR $< 1 \times 10^{-5}$), suggesting robust metabolic similarities between both analysis methods (Supplementary Fig. S8A). Using enrichment analyses, we found that at least 6/7 patients had significant enrichment for at least one of (i) enhancing tumour in bloody CSF (6/7) or (ii) non-enhancing tumour in clean CSF (3/7) (Supplementary Fig. S8B). Overall, results for the enhancing tumor microdialysate were similar to those found using paired bloody-versus-clean CSF analyses, with more robust positive enrichment scores. Less robust enrichment results for clean CSF in non-enhancing tumor microdialysate using pooled, rather than paired, analyses may be due to heterogeneity in the clean CSF metabolome between non-paired patient samples. As in Fig. 5b, enhancing versus non-enhancing metabolites from Fig. 4b demonstrated greater abundance of bloody CSF associated metabolites from the pooled analyses in enhancing tumour (Supplementary Fig. S8C). Enrichment analyses for enhancing tumour versus brain again demonstrated enrichment for bloody CSF (9/9 patients; Supplementary Fig. S9A), while non-enhancing tumour versus brain microdialysates had variable enrichment for bloody CSF (4/7 patients, Supplementary Fig. S9B), similar to the paired bloody-versus-clean CSF analyses. Collectively, these data suggest that plasma-derived metabolites from a disrupted BBB significantly contribute to the extracellular metabolome of enhancing gliomas (Fig. 5c).

One recent intraoperative study compared metabolites within the glioma's arterial versus venous supply to determine which were consumed versus produced by the gliomas[20]. We analysed their raw data maintaining thresholds we had used for our own data (FC $\geq 2$; $p \leq 0.05$), and initially found no significantly differentially abundant metabolites. By removing the FC criterion, 4 significant metabolites were identified. Taurine was more abundant in venous than arterial glioma blood, suggesting production by the tumour. In agreement with this result, we found taurine to be significantly more abundant in both enhancing and non-enhancing tumour as compared to brain

(FC E/B = 2.99x, NE/B = 2.08x). Three metabolites were more abundant in glioma arterial than venous blood, suggesting utilisation by the tumour (Supplementary Fig. S10). In agreement with our data, one of these, alanine, was part of our enhancing glioma metabolome (Table 2) and enriched for bloody CSF (Fig. 5b), consistent with the hypothesis of BBB disruption improving the glioma's access to needed metabolites. One limitation of the arterial-venous sampling study was that no parallel analyses were performed to compare metabolite utilisation and production in adjacent brain. For example, although glucose was identified as utilised by "glioma" in their data, glucose readily crosses the BBB and is highly utilised by brain, making fluorodeoxyglucose (FDG)-positron emission tomography (PET) unhelpful to discriminate tumour from brain[24]. Indeed, extracellular glucose was not differentially abundant in our tumour versus brain microdialysate. Nevertheless, these limited available data further support our hypothesis that the enhancing glioma metabolome is significantly enriched for metabolites from peripheral blood, at least some of which directly support glioma metabolism. Conversely, extracellular metabolites that are differentially abundant in both enhancing and non-enhancing tumour when compared to brain are most likely generated by tumour cells, including GAA and taurine.

## Discussion

We utilised an intraoperative window-of opportunity to perform microdialysis, sampling the extracellular metabolome of radiographically distinct regions of human gliomas in situ. Findings revealed that 1) 2-HG is robustly elevated in IDH-mutant gliomas, with potentially highest levels in non-enhancing tumour regions. 2) GAA is a previously unreported extracellular glioma metabolite that highly differentially abundant in enhancing (126.32x) and non-enhancing (13.63x) glioma as compared to brain; 3) the extracellular metabolome of enhancing glioma is conserved across patients, comprises largely amino acids and carnitines, and is enriched for plasma-derived metabolites.

Acute intraoperative microdialysis yielded relative levels of hundreds of metabolites for untargeted metabolomics analyses, including 2-HG. IDH-mutant gliomas epitomise the oncogenic relevance of altered metabolism[25,26]. We observed elevated 2-HG in the microdialysate of IDH-mutant tumours via both targeted (D vs L-2-HG) and untargeted (total D + L-2HG) analysis (Fig. 1c; Supplementary Fig. 2). Total 2-HG was previously used as a pharmacodynamic biomarker for IDH inhibitors in tissue[27] and via magnetic resonance spectroscopy[28–30]. As microdialysis can be utilised for pharmacokinetic studies of dialysable drugs[8], the ability to quantify extracellular 2-HG in vivo across a > 1000x concentration range (Fig. 1c) suggests that extracellular 2-HG within microdialysate could provide an pharmacodynamic read-out for relevant early phase clinical trials[31]. Interestingly, 2-HG was 5-12x lower in the most avidly enhancing tumour region ($n = 3$) (Fig. 1d), suggesting that some tumour metabolites may be lost down a disrupted blood brain barrier—a consideration of potential importance for longitudinal pharmacodynamic monitoring of tumour interstitial fluid.

**Table 2 . Enhancing and non-enhancing tumour metabolites.**

| Metabolite | E/B | | E/NE | | NE/B | |
|---|---|---|---|---|---|---|
| | Mean FC | *P*-value | Mean FC | *P*-value | Mean FC | *P*-value |
| Guanidinoacetate (GAA) | **126.32** | 0.0039 | 9.76 | 0.2188 | **13.63** | 0.0469 |
| proline | **10.18** | 0.0039 | **4.46** | 0.0313 | 2.09 | 0.1094 |
| deoxycarnitine | **8.71** | 0.0039 | **4.55** | 0.0156 | 1.94 | 0.3750 |
| carnitine | **7.65** | 0.0039 | **3.32** | 0.0313 | 2.18 | 0.0781 |
| acetylcarnitine (C2) | **7.47** | 0.0039 | 2.76 | 0.1094 | **2.97** | 0.0469 |
| N1-methyl-2-pyridone-5-carboxamide (2-PY) | **7.18** | 0.0391 | **3.20** | 0.0156 | 1.28 | 0.6875 |
| N6-methyllysine | **7.17** | 0.0039 | **4.67** | 0.0313 | 1.16 | 0.2969 |
| methionine | **7.09** | 0.0039 | **2.94** | 0.0156 | 2.41 | 0.0781 |
| dimethylglycine | **6.90** | 0.0039 | **3.75** | 0.0156 | 1.51 | 0.4688 |
| asparagine | **6.84** | 0.0039 | **3.64** | 0.0156 | 1.55 | 0.3750 |
| propionylcarnitine (C3) | **6.78** | 0.0078 | **3.77** | 0.0469 | 1.84 | 0.1094 |
| alanine | **6.54** | 0.0039 | **3.11** | 0.0156 | 1.96 | 0.0781 |
| valine | **6.01** | 0.0039 | **3.24** | 0.0156 | 1.70 | 0.1563 |
| tyrosine | **5.66** | 0.0039 | **3.14** | 0.0313 | 1.62 | 0.2969 |
| glycine | **5.66** | 0.0078 | 2.43 | 0.2188 | 2.18 | 0.1094 |
| imidazole lactate | **5.57** | 0.0039 | **3.78** | 0.0156 | 1.09 | 0.5781 |
| urate | **5.05** | 0.0039 | **2.63** | 0.0156 | 1.60 | 0.0781 |
| lysine | **4.54** | 0.0039 | **2.68** | 0.0156 | 1.41 | 0.2969 |
| allantoin | **4.52** | 0.0039 | **3.96** | 0.0313 | 0.90 | 0.5781 |
| histidine | **4.49** | 0.0039 | **2.30** | 0.0313 | **1.73** | 0.0156 |
| phenylalanine | **4.48** | 0.0039 | **2.59** | 0.0156 | 1.52 | 0.0781 |
| trigonelline (N′-methylnicotinate) | **4.40** | 0.0078 | 1.75 | 0.1563 | 2.57 | 0.2969 |
| 3-hydroxybutyrate (BHBA) | **4.40** | 0.0039 | **2.22** | 0.0313 | **1.77** | 0.0313 |
| 3-methyl-2-oxobutyrate | **4.34** | 0.0039 | 3.86 | 0.0781 | 0.80 | 0.5781 |
| hydroxyproline | **4.22** | 0.0039 | **2.11** | 0.0313 | 1.48 | 0.2188 |
| threonine | **4.20** | 0.0039 | **2.27** | 0.0313 | 1.58 | 0.1563 |
| leucine | **4.12** | 0.0039 | **2.43** | 0.0156 | 1.54 | 0.0781 |
| tryptophan | **3.96** | 0.0039 | **2.33** | 0.0313 | 1.40 | 0.2969 |
| N1-methyl-4-pyridone-3-carboxamide (4-PY) | 3.88 | 0.0742 | **2.16** | 0.0156 | 1.27 | 0.5781 |
| N,N,N-trimethyl-5-aminovalerate | **3.88** | 0.0039 | 2.17 | 0.0781 | 1.74 | 0.0781 |
| isoleucine | **3.76** | 0.0039 | **2.29** | 0.0156 | 1.52 | 0.0781 |
| guanine | **3.73** | 0.0078 | 1.24 | 0.5781 | **2.62** | 0.0469 |
| N,N,N-trimethyl-alanylproline betaine (TMAP) | **3.50** | 0.0039 | 1.59 | 0.1094 | 1.58 | 0.2969 |
| p-cresol sulfate | **3.36** | 0.0039 | 2.03 | 0.2969 | 1.44 | 0.2188 |
| Glutamine degradant | **3.24** | 0.0078 | **1.88** | 0.0156 | 1.61 | 0.0781 |
| arginine | **3.16** | 0.0039 | **2.17** | 0.0156 | 1.25 | 0.6875 |
| Lactate* | **3.12** | 0.0313 | 1.34 | 0.0625 | 2.32 | 0.0625 |
| ornithine | **3.07** | 0.0039 | **2.36** | 0.0313 | 1.13 | 1.0000 |
| taurine | **2.99** | 0.0117 | 1.76 | 0.1094 | **2.08** | 0.0156 |
| thioproline | **2.97** | 0.0078 | 1.31 | 0.0781 | 2.50 | 0.0781 |
| gamma-glutamylglutamine | **2.93** | 0.0117 | **1.45** | 0.0469 | **2.03** | 0.0313 |
| serine | **2.80** | 0.0039 | **2.12** | 0.0469 | 1.12 | 0.9375 |
| N6,N6,N6-trimethyllysine | **2.62** | 0.0195 | 1.80 | 0.0781 | 1.27 | 0.2188 |
| glycerophosphorylcholine (GPC) | 2.60 | 0.0742 | 0.87 | 1.0000 | **4.32** | 0.0156 |
| phenol sulfate | **2.59** | 0.0078 | 2.08 | 0.0781 | 1.09 | 0.2188 |
| N-acetylputrescine | **2.44** | 0.0273 | 1.30 | 0.8125 | 1.75 | 0.4688 |
| ergothioneine | **2.41** | 0.0391 | 0.60 | 0.3750 | **3.26** | 0.0156 |
| 3-hydroxyisobutyrate | **2.27** | 0.0391 | 1.51 | 0.3750 | 1.15 | 0.3750 |
| stachydrine | **2.24** | 0.0078 | 1.45 | 0.2188 | 1.39 | 0.0781 |
| 1-methyl-5-imidazolelactate | 2.13 | 0.1073 | 0.56 | 0.8125 | **4.27** | 0.0156 |
| 7-methylguanine | **2.10** | 0.0195 | 1.53 | 0.3750 | 1.40 | 0.2945 |
| N-acetylglycine | **2.05** | 0.0039 | 1.71 | 0.1563 | 1.10 | 0.2188 |
| glycerophosphoglycerol | 2.04 | 0.2340 | 0.45 | 0.4688 | **5.21** | 0.0156 |
| malate | 1.87 | 0.1641 | 0.70 | 0.9375 | **2.99** | 0.0156 |
| phosphoethanolamine (PE) | 1.56 | 0.6523 | 0.49 | 0.5781 | **4.83** | 0.0156 |
| creatine | 1.05 | 0.5703 | 0.55 | 0.2188 | **2.25** | 0.0313 |
| glycerol-3-phosphate | 1.02 | 0.5703 | 0.40 | 0.3750 | **3.25** | 0.0313 |
| phosphate | 0.87 | 1.0000 | **0.33** | 0.0313 | **2.91** | 0.0469 |
| glycerophosphoethanolamine | 0.82 | 0.8203 | 0.43 | 0.1563 | **2.83** | 0.0156 |
| inosine | 0.64 | 0.4258 | **0.48** | 0.0156 | 1.60 | 0.1563 |
| myo-inositol | 0.51 | 0.4258 | **0.46** | 0.0469 | 1.13 | 0.5781 |
| orotate | 0.47 | 0.1641 | **0.39** | 0.0313 | 0.98 | 0.8125 |
| N-formylmethionine | 0.42 | 0.1289 | **0.36** | 0.0469 | 1.13 | 0.6875 |
| aspartate | **0.40** | 0.0195 | 0.43 | 0.2188 | 1.06 | 0.6875 |

**Table 2 (continued)**

| Metabolite | E/B | | E/NE | | NE/B | |
|---|---|---|---|---|---|---|
| | Mean FC | P-value | Mean FC | P-value | Mean FC | P-value |
| lyxonate | **0.39** | 0.0078 | 0.47 | 0.0781 | 0.86 | 0.9375 |
| N-acetylasparagine | **0.37** | 0.0039 | **0.37** | 0.0469 | 0.86 | 0.6875 |
| 3-hydroxy-3-methylglutarate | **0.37** | 0.0391 | 0.29 | 0.0781 | 1.55 | 0.1563 |
| N-acetylglutamine | **0.34** | 0.0078 | **0.34** | 0.0156 | 0.99 | 0.6875 |
| N-acetylthreonine | **0.34** | 0.0078 | **0.34** | 0.0469 | 0.93 | 1.0000 |
| N-acetyl-beta-alanine | **0.30** | 0.0391 | 0.45 | 0.1422 | 0.73 | 0.6875 |
| N-acetylalanine | **0.30** | 0.0391 | **0.25** | 0.0156 | 0.96 | 0.9375 |
| 2-methylcitrate/homocitrate | **0.29** | 0.0039 | **0.27** | 0.0156 | 0.94 | 0.6875 |
| N-acetylneuraminate | **0.25** | 0.0039 | 0.32 | 0.0781 | 0.75 | 0.9375 |
| fructose | **0.25** | 0.0039 | **0.40** | 0.0469 | 0.56 | 0.4688 |
| N1-methylinosine | **0.23** | 0.0078 | **0.20** | 0.0313 | 1.09 | 0.2969 |
| N-acetylserine | **0.21** | 0.0039 | **0.23** | 0.0313 | 0.87 | 0.6875 |
| Erythronate | **0.18** | 0.0039 | **0.21** | 0.0156 | 0.83 | 1.0000 |
| N-acetylaspartate (NAA) | **0.17** | 0.0039 | **0.16** | 0.0156 | 1.18 | 0.5781 |
| N-acetyl-aspartyl-glutamate (NAAG) | **0.16** | 0.0078 | 0.19 | 0.1563 | 0.73 | 0.6875 |
| 4-acetamidobutanoate | **0.15** | 0.0273 | 0.28 | 0.1563 | 0.96 | 0.9375 |
| 2,3-dihydroxy-5-methylthio-4-pentenoate (DMTPA) | **0.15** | 0.0039 | **0.24** | 0.0156 | 0.55 | 0.0781 |
| 2-O-methylascorbic acid (2-O-MA) | **0.14** | 0.0039 | **0.15** | 0.0156 | 0.83 | 0.8125 |
| ribonate | **0.11** | 0.0039 | **0.13** | 0.0156 | 0.82 | 0.9375 |
| arabonate/xylonate | **0.08** | 0.0039 | **0.14** | 0.0156 | 0.70 | 0.4688 |
| arabitol/xylitol | **0.04** | 0.0039 | **0.08** | 0.0156 | 0.57 | 0.3750 |

Significantly altered metabolites across enhancing or non-enhancing tumour and brain microenvironments in paired patient specimens based on Wilcoxon rank-sum test for non-parametric distributions ($n = 9$ for E/B and $n = 7$ for E/NE or NE/B; means of fold-changes (FC)), from Fig. 4. *bold = $p < 0.05$. See Supplemental Data, Fig. 4 tab for all metabolites' fold-changes and $p$-values.
aFold-change and $p$-value for lactate are based on $n = 7$ cases, rather than $n = 9$, as the perfusate solution contained lactate in 2 cases.

Importantly, although *IDH*-mutations are thought to drive early tumourigenic metabolic and epigenetic changes[26,32], the global extracellular metabolomes of enhancing *IDH*-mutant versus *IDH*-WT grade 4 astrocytomas were otherwise indistinguishable (Figs. 2a, 3a). Recently published work from over 200 glioma tissues demonstrated less separation between *IDH*-mutant grade 4 astrocytomas from GBMs[21] than lower grade *IDH*-mutant gliomas, potentially suggesting metabolic convergence of high-grade gliomas. Greatest similarities were seen across patients in the extracellular metabolome of contrast-enhancing tumours, which could be expected if improved availability of plasma-associated metabolites impacts tumour metabolism.

Cancer-associated metabolic vulnerabilities are of increasing interest as therapeutic targets. In contrast to gliomas' notorious genomic instability[33,34], a more focused inventory of metabolic strategies may be utilised to combat bioenergetic demands[35], oxidative stress[36], hypoxia[37], and nutrient deprivation[38,39]. Our rank-based analyses revealed an enhancing glioma extracellular metabolome that was consistent across primary and recurrent *IDH*-WT and mutant lesions, as well as an H3K27M-mutant tumour (Fig. 3a(i, ii)). This signature appeared comparable to that previously obtained in the context of intra-or-post-operative microdialysis[12–16], consistent with reproducibility of the enhancing glioma metabolome across clinical and technical variables (Supplementary Table 1), research teams and contexts spanning intra-operative versus post-operative sampling. However, our data demonstrate for the first time, to our knowledge, that most of the metabolites in the enhancing glioma extracellular metabolome are absent from the non-enhancing glioma metabolome. These results suggest that the extracellular metabolome of enhancing gliomas may be largely defined by metabolites from systemic circulation that more easily access the tumour microenvironment through a disrupted BBB. Although catheter insertion may hypothetically result in some degree of BBB disruption, intra-patient catheter comparisons revealed significant enrichment for plasma-associated metabolites, suggesting that the extent BBB disruption induced from catheter insertion was negligible compared to the tumour-induced disruption in enhancing versus non-enhancing tumour. Furthermore, our findings are in alignment with the observation that aggressive portions of gliomas avidly uptake amino acids from the systemic circulation, serving as the basis for highly sensitive and specific glioma PET tracers, including tyrosine[40] and methionine[41], both of which were more abundant in enhancing glioma than non-enhancing tumour or brain (Fig. 4; Table 2).

Given the established importance of multiple identified plasma- and enhancing tumour-associated metabolites, including amino acids and carnitines to cancer biology[42–44], BBB disruption itself may underlie the metabolic similarities across molecularly diverse HGGs by providing a continuous source of protumourigenic metabolites. Anecdotally, our patient with a completely non-enhancing molecular GBM had evidence of this lesion on a CT obtained 5 years prior to diagnosis—consistent with a lower rate of growth than patients with enhancing lesions. Indeed, the adverse prognostic impact of a contrast-enhancement in gliomas, including histologically low-grade tumours, is well recognised[45,46], as is the positive impact on survival of gross total resection of enhancing tumour across all molecularly defined GBM subgroups and ages[47]. As such, our findings suggest that blood brain barrier disruption itself may transform the extracellular glioma metabolome into a wellspring of nutrients for accelerated growth, particularly in enhancing tumour regions.

At this point, we cannot definitively answer the "chicken" or "egg" question of whether BBB disruption occurs for the purpose of accessing systemically-derived analytes or if it is simply a pre-existing process that a glioma can utilise to its advantage. Many of the plasma-associated metabolites enriched in enhancing tumour have been reproducibly associated with cancer biology[48], including the amino acids proline[42], glycine[49], and amino acid derivatives, such as N6-methyllysine. Similarly, carnitines (Fig. 4a, b, diii, iv) have been linked to energy production via fatty acid oxidation in glioma[50,51]. It is possible that the induction of

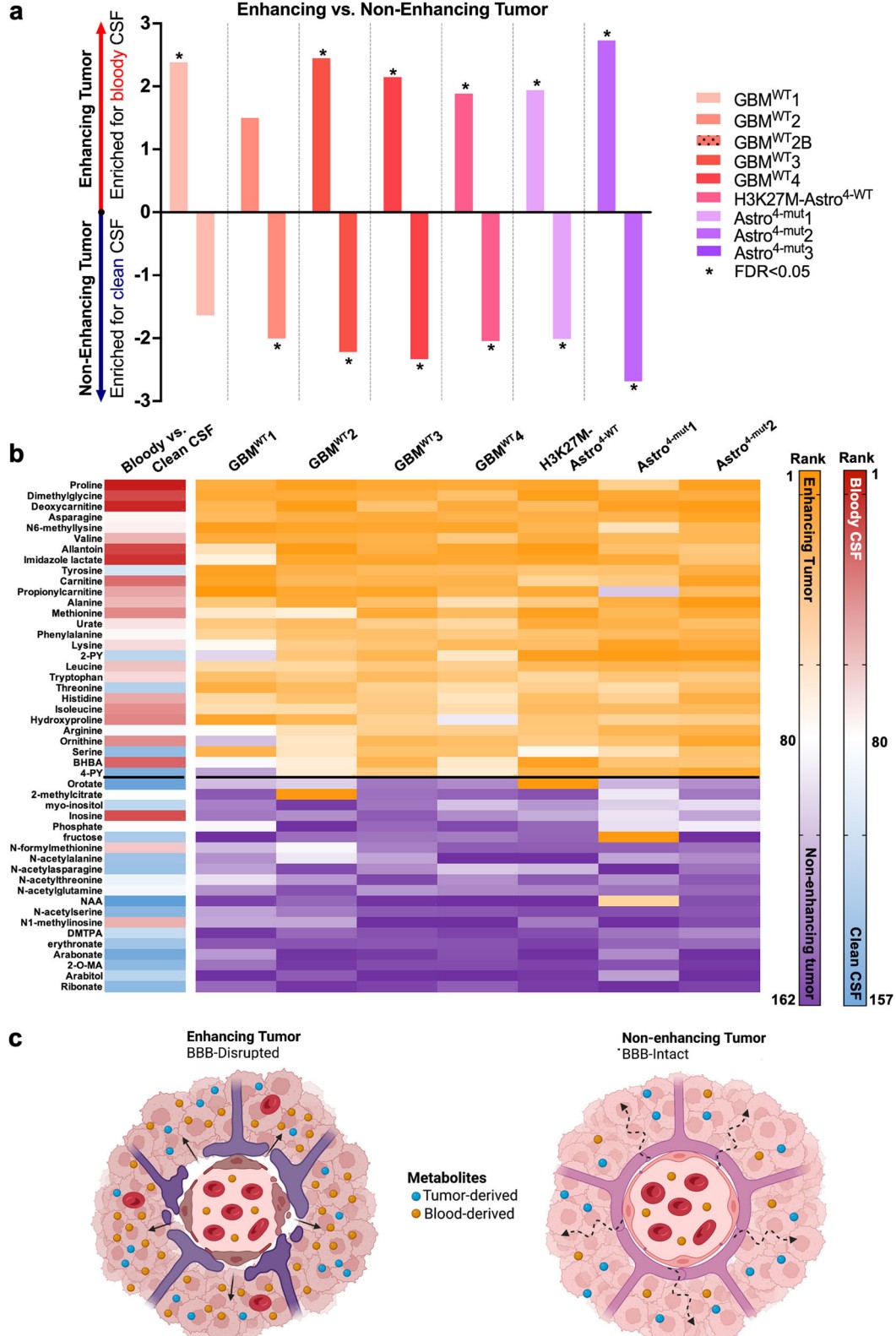

**Fig. 5 Enhancing glioma microdialysate is enriched for plasma-derived metabolites. a** Enrichment analysis was utilised to determine the enrichment of each patient's enhancing vs. non-enhancing tumour catheters (ranked list: Cath. X vs. Y) for high-plasma versus low-plasma CSF (metabolite sets). Positive normalised enrichment scores (NES) indicate metabolic similarities to high-plasma CSF; negative NES indicate metabolic similarities to low-plasma CSF (* = FDR ≤ 0.05). **b** The rank of each metabolite from the significant enhancing versus non-enhancing glioma metabolite in Fig. 4b is shown for each patient (E: purple, rank 1, to NE: orange, rank 162), along with the metabolite's rank in a bloody vs. clean CSF ranked fold-change list (bloody: red, 1 – clean: blue, 157). **c** Proposed model depicting the impact of blood-brain barrier disruption on the enhancing versus non-enhancing glioma extracellular metabolomes (created using BioRender).

blood-brain barrier disruption by proliferating gliomas[52] may represent an adaptive response to nutrient deprivation observed in gliomas[53–55], catalysing a feedforward cycle of nutrient utilisation and recruitment. Mechanistic studies selectively manipulating BBB permeability or extracellular plasma abundance will be required to dissect these interactions. If our hypothesis of disrupted BBB-mediated tumour aggressiveness is correct, maintaining and restoring BBB integrity may prove an independently important therapeutic goal, distinct from angiogenesis-blocking agents such as bevacizumab[56].

Although we used UPLC-MS/MS to maximise our yield of potentially novel glioma metabolites, we had not expected to find a new extracellular glioma metabolite that could outperform all other enhancing glioma metabolites by over 10-fold. Although GAA was present at a low abundance in bloody CSF and not detected in clean CSF, its disproportionately high fold change in both enhancing and non-enhancing tumour versus brain (126.32x, and 13.63x, respectively; Fig. 4a, d(i), Table 2) suggest it may be produced by the tumour. Of note, while a small portion of GAA in enhancing tumor may be plasma-derived, 2-HG is uniquely aberrantly created by IDH-mutant tumors. There would thus be a less profound tumor-versus-plasma concentration gradient for GAA as compared to 2-HG, perhaps explaining why 2-HG, but not GAA, was more elevated in non-enhancing than enhancing tumour microdialysate. No CNS efflux transporter exists for GAA at the BBB[57]. As such, if generated as a result of glioma metabolism, it is liable to accumulate. GAA also accumulates in patients with GAMT deficiency, leading to impaired creatine production, seizures, and cognitive impairment[58]. GAA is a precursor for creatine, levels of which were slightly elevated in non-enhancing (2.25x; $p = 0.03$), but not enhancing tumour (Table 2). Interestingly, GAA is co-produced with ornithine, which was slightly higher in enhancing tumour when compared to non-enhancing tumour (2.36x, $p = 0.03$) and brain (3.07x, $p = 0.004$) (Table 2). Ornithine is the substrate for polyamine synthesis via ornithine decarboxylase (ODC) which is upregulated in cancers, including glioblastoma, to support catabolic demand, regulate intracellular pH in the acidic tumour core, and protect against immune surveillance[59]. It is therefore tempting to speculate that increased GAA could be a byproduct of increased glioma polyamine synthesis and utilisation. If true and based on the substantial fold-change, extracellular GAA could serve as a biomarker of not only local glioma abundance but also pharmacodynamic response to ODC blockade. ODC blockade with Difluoromethylornithine (DFMO) remains in clinical trials for glioma (NCT02796261); though upregulation of polyamine transporters may mediate metabolic compensation leading to improved survival with dual blockade of ODC and polyamine transporters (via AMXT-1501), in preclinical models of diffuse midline gliomas and other cancers[60,61]. To determine if GAA can be used as a dynamic biomarker of ODC production in situ, we are opening a post-operative microdialysis study under an IND, monitoring microdialysate levels of GAA in unresectable regions of residual tumour with or without DFMO +/− AMXT1501[62]. Further dynamic insights may be achievable with isotope tracing and are under development.

With the help of our patients and intraoperative microdialysis, we have shown that GAA is surprisingly abundant in the glioma microenvironment, and that plasma-associated metabolites largely define the enhancing glioma microenvironment, potentially positioning them to support glioma aggressiveness. We acknowledge the limitations of this study, including the small sample size of fourteen patients across fifteen surgeries, which may limit the generalizability of our results despite robust metabolic similarities across two heterogeneous cohorts of patients. Despite this small sample size, given the urgent need for more in situ studies from patients, we present their data as a first

installment toward what we intend to be a growing collaborative, open-source repository of individualised data from live human gliomas. This intra-operative microdialysis trial remains open and we are continuing to accumulate more data to evaluate the reproducibility of our results in a larger cohort of diverse patients, including those with low-grade gliomas and more molecularly heterogeneous high-grade gliomas. Our patients understand the formidable odds of their disease. We (both patients and investigators) hope that the data and insights shared can help inspire patient-centric translational paradigms. By leveraging neurosurgical access to live human disease, we aim to accelerate tangible progress within the lifetimes of individual patients.

## Methods

**Patient cohort, study design, and intraoperative microdialysis.** All study procedures were approved by the Mayo Clinic Institutional Review Board (IRB) and performed according to relevant ethical regulations. Patients provided written informed consent to participate in NCT04047264--an ongoing study utilising intraoperative high molecular weight (100 kDa) microdialysis under an investigational device exemption. Study eligibility included adults ( > 18yo) undergoing a clinically indicated surgery for known or suspected glioma. Each patient ($n = 14$; 15 surgeries; Table 1) underwent intraoperative microdialysis using up to three 100 kDA catheters and variable rate microdialysis pumps (M Dialysis 71 High Cut-Off Brain Microdialysis Catheters and 107 Microdialysis Pump, respectively) across radiographically diverse regions (Fig. 1a, b). Microdialysis was performed at 2 µL/min based on a prior intra-operative study demonstrating that this flowrate could enable sufficient collection of analytes for mass spectrometry[14], with collection vials exchanged every 20 min. See Supplementary methods for procedural details, pathology, and molecular tumour analyses. Each patient is referred to via a unique de-identified label reflecting their histology, IDH-status, and WHO grade according to 2021 WHO classification[63] (Table 1). The study was initially performed in a pilot cohort of patients (GBM^WT1, GBM^WT2, Astro^4-mut1, Oligo^21, and Oligo^31) and the sample size then expanded once the safety and feasibility of the study had been demonstrated.

**Targeted analysis of D/L-2-HG.** Targeted metabolomic analysis of microdialysate was performed by the Mayo Clinic Metabolomic Core facility. L and D isomers of 2-hydroxyglutaratic acid were derivatised and quantified by liquid chromatography mass spectrometry (LC/MS) using slight modifications to previously described methods[64–66]. See supplementary methods for details.

**Untargeted metabolomic analysis.** Untargeted metabolomic analysis was performed by Metabolon, Inc. Ultra-performance liquid chromatography tandem mass spectrometry (UPLC-MS/MS), which combines physical separation of liquid chromatography with the mass analysis capabilities of mass spectrometry. See supplementary methods for details.

**Ranked metabolite lists.** Ranked metabolite lists of enhancing or non-enhancing tumour (Catheter X or Y) versus brain (Catheter Z) were generated from fifteen cases based on calculated fold changes of the peak area (e.g., X/Z or Y/Z) for each metabolite. A tumour versus brain ranked metabolite list could not be generated for one patient (Oligo^2) given all catheters were located within tumour. In patients with both enhancing and non-enhancing tumour catheters ($n = 7$ patients), a ranked metabolite lists of enhancing vs. non-enhancing tumour was generated (Catheter X/Y). To understand the potential impact of extracellular blood-derived metabolites on microdialysate metabolic signatures, we additionally generated a ranked list of bloodiness-associated metabolites using a pair of cranial CSF samples obtained from a single patient prior to resection of a 4th ventricular ependymoma. Both samples were obtained within approximately 1 min of each other—one with and one without blood contamination from the surgical wound. Twenty-six other intracranial CSF samples were collected with variable blood-contamination. Samples were centrifuged at 400 Gy for 10 min to remove red blood cells and other debris and aliquoted. For the paired bloody and clean CSF sample, the ranked metabolite list was based on metabolite fold-change (bloody/non-bloody) between samples, after filtering for metabolites present in both CSF and > 90% of microdialysate samples. For the pooled bloody and clean CSF samples, CSF samples were ranked from greatest to least blood contamination based on heme content. Excluding the ependymoma sample, CSF from the top (bloody) and bottom (clean) quartiles were utilized ($n = 7$ each), filtering for metabolites present > 90% of microdialysate samples and > 85% of CSF samples. Ranked metabolite lists were then generated based on the metabolite fold-change for the average of the bloody CSF samples versus the average of the clean CSF samples. See supplementary methods for details on enrichment ranked lists analyses.

**Enrichment analysis.** Gene Set Enrichment Analysis (GSEA) is the most widely utilised analytical method for rank-based analysis of data sets in biomedicine.

GSEA is frequently performed using curated gene sets from the molecular signatures database (MSigDB), though custom analyte sets can be used, enabling enrichment analysis of metabolites rather than genes. In GSEA, custom analyte sets are queried against a ranked list of analytes to determine where those analytes fall on the ranked list (Supplementary Fig. S1). If the analytes are most frequently found at the top of the ranked list, this is known as "positive enrichment;" if they fall at the end of the ranked list, this is known as "negative enrichment." The software uses this information to calculate a normalised enrichment score and a p-value. FDR values are also provided to correct for multiple hypothesis testing when multiple analyte sets are evaluated simultaneously for enrichment in a particular data set. Custom metabolite libraries were created using the top and bottom 35 metabolites from each ranked metabolite list described above and can be found in Supplementary Data (.gmx file). See supplementary methods for enrichment analysis details.

**MetaboAnalyst**. Metabolomic analyses were performed using MetaboAnalyst (5.0), a publicly available web-based tool to analyse and visualise metabolomic data[67]. The normalised peak area data for the metabolites present across at least 40/44 catheters were entered into MetaboAnalyst without further filtering. Assuming a monotonical rather than linear relation between metabolites and samples, Spearman Correlation maps were generated in MetaboAnalyst based on features (metabolites present across all samples) and samples.

**Statistics and reproducibility**. Data are presented as normalised peak areas (median = 1 for each metabolite within each analysed batch. Raw and normalised peak area data are provided in Supplementary Data. MetaboAnalyst 5.0 was used for Spearman correlations. Enrichment Analysis was performed using GSEA 4.1.0 (Gene Set Enrichment Analysis), repurposed for metabolite set analysis using custom metabolite sets. The normal distribution of metabolites in Fig. 4 and Table 2 was tested via D'Agostino-Pearson test; a Wilcoxon signed rank test for non-parametric distributions was then performed on paired enhancing tumour and brain metabolites using GraphPad PRISM 9.1. Volcano plot cut-offs were set at $FC \geq 2$ and $p \leq 0.05$. Graphs were generated using GraphPad PRISM 9.1. FDR $\leq 0.05$ was considered statistically significant for enrichment analysis. FDR values given as 0 by GSEA were reported as $<1 \times 10^{-5}$.

**Reporting summary**. Further information on research design is available in the Nature Portfolio Reporting Summary linked to this article.

## Data availability

All data from this study are provided as a supplementary data file. Source data for the graphs and charts in the figures are also available in a supplementary data file. Any remaining information can be obtained from the corresponding author upon reasonable request.

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

## Acknowledgements

We would like to thank each of our patients who selflessly participated in this study and without whom this work and any advancement in the field of neuro-oncology would not be possible. We also thank Elizabeth Oishi for regulatory support and the Mayo Clinic Neurosurgery Clinical Research team, and the Mayo neurosurgery operative staff for their invaluable support. CRC was supported by the National Institute of Health T32GM065841. TCB was supported by NINDS NRCDP K12, NINDS R61 NS122096, Mayo Clinic Center for Individualized Medicine and CCaTS award UL1TR002377, the American Brain Tumor Association, Brains Together for the Cure, Humor to fight the Tumor, and Lucius & Terrie McKelvey. Support was provided to TCB, SHK, AEW, and RAV through NCI R37CA276851. Support was also provided through NCI U54 CA210180 (JS).

## Author contributions

Study design and conception: T.C.B., C.R.C., L.P.C., K.R., W.F.E. - Performed experiments: T.C.B., C.R.C., L.P.C., K.R., K.J.M., J.J.W., D.A.B., B.T.H., I.J.T. - Analysed data: T.C.B., C.R.C., K.R.. - Wrote the manuscript: C.R.C., T.C.B. - Critically reviewed manuscript: C.R.C., L.P.C., K.R., A.M.C., M.Ra., A.G.L., D.A.B., K.J.M., J.J.W., B.T.H., I.J.T., S.I., S.C.R., R.H., J.O., W.F.E., J.N.S., R.A.V., M.Ro., A.E.W., S.H.K., T.C.B.

## Competing interests

The authors declare no competing interests.

## Ethics approval and consent to participate

This study was approved by the Mayo Clinic Institutional Review Board and all participants provided their consent to participate in this study. This study was performed in accordance with the Declaration of Helsinki.

## Consent for publication

All participants have provided consent for publication.
