## [Peer Review File · Communications Biology]

Reviewers' comments:

Reviewer #1 (Remarks to the Author):

This paper examines the extracellular microenvironment of gliomas and how it modulates cancer behavior. The authors used intra-operative microdialysis to sample the extracellular metabolome of different regions during neurosurgical resections. The authors suggest metabolite diffusion through a disrupted blood-brain barrier may largely define the enhancing extracellular glioma metabolome. They also found that the enhancing but not the non-enhancing glioma metabolome was significantly enriched for plasma-associated metabolites comprising amino acids and carnitines. This paper provides valuable insights into the extracellular microenvironment of gliomas and how it modulates cancer behavior.

While the findings are interesting and provide valuable insights into the extracellular microenvironment of gliomas, it is important to acknowledge that the small sample size may limit the generalizability of the results. A larger sample size would allow for more robust statistical analysis and may help confirm this study's findings. However, it is also worth noting that obtaining intra-operative microdialysis samples is a technically challenging and invasive procedure, which may have contributed to the small sample size.

The authors only analyzed samples from a single cohort of patients, which may limit the generalizability of the results. Additionally, the patients in this study were undergoing neurosurgical resection for glioma, which may not represent the broader population of glioma patients. To validate the findings of this study and ensure that they can be generalized to other populations, it may be necessary to include patients from multiple cohorts in future studies. A validation cohort would provide an opportunity to confirm the findings and determine whether they are consistent across different patient populations.

Overall, while the findings of this study are important and provide valuable insights into the extracellular microenvironment of gliomas, it is important to acknowledge the limitations of the study, including the small sample size, lack of a validation cohort, and technical challenges associated with obtaining intra-operative microdialysis samples.

Reviewer #2 (Remarks to the Author):

Summary of the key results

The authors use a microdialysis apparatus to intraoperatively collect metabolites in the glioma microenvironments for global metabolite profiling. They reported that contrast-enhancing (radiographically blood-brain barrier (BBB) disrupted) regions of high-grade gliomas (HGGs) exhibited a conserved extracellular metabolome with a characteristic of blood-enriched metabolites. The authors suggested that 2-HG diffusion to blood in the enhancing region may explain a lower 2-HG concentration in enhancing region than non-enhancing region. The global metabolite profiling for the glioma microenvironments adds experimental evidence to the feedforward cycle of gliomas disrupting the BBB and utilizing more nutrients from blood for tumor growth, suggesting BBB repair could become one of the important goals of treatment. In addition, the authors identified the metabolite guanidinoacetate (GAA) was enriched consistently enriched in both enhancing and non-enhancing gliomas tumor, which may become a useful biomarker for clinical studies.

Comments:

1. Overall, the analysis approaches are appropriate and the analyzed results support the main conclusions. One small caveat is that only one bloody CSF and one clean CSF are used to establish the blood metabolite set for GSEA analysis. It is understandable because these samples are very difficult to obtain. One suggestion is to use literature information to build a blood metabolite set.
2. The discovery of GAA being a differential metabolite is an important finding for the manuscript. The authors mention GAMT deficiency could result to GAA accumulation. It would be great to include patient data for GAMT deficiency, or to discuss GAMT's role in the current finding.

Others:

P3L18. Cited ref 13-15 are more than ten years from now. Citing more recent publications on the related topic may further highlight the technical and biological advancement in the current manuscript. P7L23-P8L12. Some of the sentences may go to the introduction or method section so that the results could stand out better.

P9L26. Is elevated 2HG in blood a biomarker for more invasive IDH mutant gliomas? If literatures support this idea, it can support that 2HG level high in non-enhancing region than enhancing region is due to the 2HG diffusion to the disrupted blood brain barrier.

P13L5 & L16. $FDR = 0.000$ or $FDRs = 0$ may be the results of software omitting digits after a few zeros. FDR could be very small, such as 10^{-6} , but should not be 0. Consider to replace the expression with FDR below a certain threshold.

Minor typos in P9L25 and P16L15.

Response to Reviewer Comments

Reviewers' comments:

Reviewer #1 (Remarks to the Author):

This paper examines the extracellular microenvironment of gliomas and how it modulates cancer behavior. The authors used intra-operative microdialysis to sample the extracellular metabolome of different regions during neurosurgical resections. The authors suggest metabolite diffusion through a disrupted blood-brain barrier may largely define the enhancing extracellular glioma metabolome. They also found that the enhancing but not the non-enhancing glioma metabolome was significantly enriched for plasma-associated metabolites comprising amino acids and carnitines. This paper provides valuable insights into the extracellular microenvironment of gliomas and how it modulates cancer behavior.

While the findings are interesting and provide valuable insights into the extracellular microenvironment of gliomas, it is important to acknowledge that the small sample size may limit the generalizability of the results. A larger sample size would allow for more robust statistical analysis and may help confirm this study's findings. However, it is also worth noting that obtaining intra-operative microdialysis samples is a technically challenging and invasive procedure, which may have contributed to the small sample size.

The authors only analyzed samples from a single cohort of patients, which may limit the generalizability of the results. Additionally, the patients in this study were undergoing neurosurgical resection for glioma, which may not represent the broader population of glioma patients. To validate the findings of this study and ensure that they can be generalized to other populations, it may be necessary to include patients from multiple cohorts in future studies. A validation cohort would provide an opportunity to confirm the findings and determine whether they are consistent across different patient populations.

Overall, while the findings of this study are important and provide valuable insights into the extracellular microenvironment of gliomas, it is important to acknowledge the limitations of the study, including the small sample size (comment #1), lack of a validation cohort (comment #2), and technical challenges associated with obtaining intra-operative microdialysis samples (comment #3).

Author response to reviewer #1:

We thank the reviewer for their positive review of our manuscript, including its interesting findings and “valuable insights into the extracellular microenvironment of gliomas and how it modulates cancer behavior.” We also thank the reviewer for their helpful feedback and suggestions on how the manuscript could be improved. Please see the table below for our response to the reviewer’s comments and changes made in the manuscript to address these comments.

Reviewer comment	Response	Manuscript
1. “It is important to acknowledge that the small sample size may limit the generalizability of the results. A larger sample size would allow for more robust statistical analysis and may	We agree with the reviewer’s assessment of our small sample size and the impact thereof on the generalizability of the results. While we have identified a robust set of metabolic similarities in the enhancing, but not non-	Addition: “We acknowledge the limitations of this study, including a small sample size of fourteen patients across fifteen surgeries, which may limit the generalizability of our results despite robust metabolic similarities across

Response to Reviewer Comments

help confirm this study's findings.”	enhancing portions of patient's tumors, often at very low FDRs, we acknowledge that these results will need to be validated in a larger cohort of patients. The trial remains open and we are continuing to accumulate more data to evaluate the reproducibility of our study's findings, in addition to any metabolic differences that may not have yet been identified in our cohort. We have edited the manuscript as described in the next column to reflect these changes.	two heterogeneous cohorts of patients.” (Pages 17-18; lines 33-34,1) Addition: “This intra-operative microdialysis trial remains open and we are continuing to accumulate more data to evaluate the reproducibility of our results in a larger cohort of diverse patients, including those with low-grade gliomas and more molecularly heterogeneous high-grade gliomas.” (Page 18, lines 7-10)
2. “The authors only analyzed samples from a single cohort of patients, which may limit the generalizability of the results. Additionally, the patients in this study were undergoing neurosurgical resection for glioma, which may not represent the broader population of glioma patients. To validate the findings of this study and ensure that they can be generalized to other populations, it may be necessary to include patients from multiple cohorts in future studies. A validation cohort would provide an opportunity to confirm the findings and determine whether they are consistent across different patient populations.”	We thank the reviewer for this very important comment and apologize for our lack of clarity. This study was indeed performed in two different cohorts. Results were initially identified in a first cohort of three patients with enhancing high-grade gliomas from our pilot study (GBM^{WT1}, GBM^{WT2}, and Astro^{4-mut1}). These results were then independently evaluated in a second cohort of patients, including one patient from the first cohort's repeat surgery (GBM^{WT2B}; others: GBM^{WT3}, GBM^{WT4}, H3K27M-Astro^{4-WT}, Astro^{4-mut2}, and Astro^{4-mut3}). The second cohort of patients included a more diverse cohort of patients, particularly in regard to resection of recurrent gliomas (3 patients) and an H3K27M-mutated patient. Despite this, as seen in Supplemental Figure S4, we could reproducibly identify the enhancing glioma signature in the second cohort of patient when the metabolites were ranked according based on the first cohort of patients. The results	Please see after this table for the relevant supplemental figure with the two cohorts of patients. Addition: “The study was initially performed in a pilot cohort of patients (GBM^{WT1}, GBM^{WT2}, Astro^{4-mut1}, Oligo², and Oligo³¹) and the sample size then expanded once the safety and feasibility of the study had been demonstrated.” (Page 5, lines 17-20) Addition: “Although the magnitude of these fold changes varied between patients (Supplementary Fig. S3), the rank of enhancing tumour versus brain-associated metabolites appeared similar in 3/3 patients with enhancing tumours from our initial cohort (Supplementary Fig. S4A, first and second cohort).” (Page 10, lines 15-18) Addition: “Additionally, paired enhancing versus non-enhancing tumour catheters were available from 7

Response to Reviewer Comments

	are described in the section, “A reproducible metabolome of enhancing glioma versus brain” on page 10, lines 1-26. To clarify that there were two cohorts of analyses performed, we have made changes to the manuscript as described in the next column.	patients. Similar rank-based analyses were performed for enhancing versus non-enhancing tumour which revealed a surprisingly conserved rank of enhancing-to-non-enhancing metabolites across patients (Fig. 3Bii; results presented as first and second cohort in Supplementary Fig. S4B).” (Page 10, lines 22-26) Addition: “Conversely, the distribution of non-enhancing catheter-associated metabolites appeared less consistent (Fig. 3Ci; results presented as first and second cohort in Supplementary Fig. S4C).” (Page 10, lines 26-28)
3. “However, it is also worth noting that obtaining intra-operative microdialysis samples is a technically challenging and invasive procedure, which may have contributed to the small sample size.” “Overall, while the findings of this study are important and provide valuable insights into the extracellular microenvironment of gliomas, it is important to acknowledge the limitations of the study, including... technical challenges associated with obtaining intra-operative microdialysis samples.”	As commented on by the reviewer, there are certain aspects of intra-operative microdialysis which may be technically challenging at first, particularly in regard to the pre-operative planning required to maximize the sampling time while the remainder of the resection is ongoing. However, we have generally found that after a few cases, microdialysis becomes quite feasible to deploy while the resection is ongoing, especially with larger lesions and cases with speech/motor mapping, where sampling can occur while baselines are obtained for intra-operative mapping. Our reasoning for publishing these results, despite an admittedly small sample size, is that we believe there is an urgent need to provide more in situ data from live human gliomas within our patients’ lifetimes. Toward this goal, we provide our patients’ complete	Edit: “Despite this small sample size, given the urgent need for more in situ studies from patients, we present their data as a first installment toward what we intend to be a growing collaborative, open-source repository of individualised data from live human gliomas.” (Page 18, lines 2-5)

Response to Reviewer Comments

	metabolic data as part of this manuscript in order to maximize and accelerate the number of valuable discoveries which can be performed. We have clarified this rationale in the manuscript.	
--	---	--

Supplemental figure S4:

Supplementary Figure S4. Metabolic signatures of the enhancing versus non-enhancing glioma versus brain metabolites, using first and second cohort-based analyses.

162 metabolites present in at least 40/44 catheters were ranked according to the (A) enhancing tumour-versus-brain (X/Z), (B) enhancing versus non-enhancing tumour (X/Y), or (C) non-enhancing tumour versus brain (Y/Z) fold change in each patient. The rank order of each metabolite in each 2-catheter tumour/brain comparison (e.g., catheter X versus Z) is conveyed as a heat map from 1 to 162 (orange: enhancing tumour, purple: non-enhancing tumour, brain: green). Metabolites are listed based on the average of ranks of (A) enhancing vs. brain (B) enhancing vs. non-enhancing tumour, and (C) non-enhancing tumour vs. brain in the first cohort (Astro^{4-mut1}, GBM^{WT1} and GBM^{WT2}).

Response to Reviewer Comments

Reviewer #2 (Remarks to the Author):

Summary of the key results

The authors use a microdialysis apparatus to intraoperatively collect metabolites in the glioma microenvironments for global metabolite profiling. They reported that contrast-enhancing (radiographically blood-brain barrier (BBB) disrupted) regions of high-grade gliomas (HGGs) exhibited a conserved extracellular metabolome with a characteristic of blood-enriched metabolites. The authors suggested that 2-HG diffusion to blood in the enhancing region may explain a lower 2-HG concentration in enhancing region than non-enhancing region. The global metabolite profiling for the glioma microenvironments adds experimental evidence to the feedforward cycle of gliomas disrupting the BBB and utilizing more nutrients from blood for tumor growth, suggesting BBB repair could become one of the important goals of treatment. In addition, the authors identified the metabolite guanidinoacetate (GAA) was enriched consistently enriched in both enhancing and non-enhancing gliomas tumor, which may become a useful biomarker for clinical studies.

Comments:

1. Overall, the analysis approaches are appropriate and the analyzed results support the main conclusions. One small caveat is that only one bloody CSF and one clean CSF are used to establish the blood metabolite set for GSEA analysis. It is understandable because these samples are very difficult to obtain. One suggestion is to use literature information to build a blood metabolite set.
2. The discovery of GAA being a differential metabolite is an important finding for the manuscript. The authors mention GAMT deficiency could result to GAA accumulation. It would be great to include patient data for GAMT deficiency, or to discuss GAMT's role in the current finding.

Others:

P3L18. Cited ref 13-15 are more than ten years from now. Citing more recent publications on the related topic may further highlight the technical and biological advancement in the current manuscript. (Comment #3)

P7L23-P8L12. Some of the sentences may go to the introduction or method section so that the results could stand out better. (Comment #4)

P9L26. Is elevated 2HG in blood a biomarker for more invasive IDH mutant gliomas? If literatures support this idea, it can support that 2HG level high in non-enhancing region than enhancing region is due to the 2HG diffusion to the disrupted blood brain barrier. (Comment #5)

P13L5 & L16. FDR = 0.000 or FDRs = 0 may be the results of software omitting digits after a few zeros. FDR could be very small, such as 10⁻⁶, but should not be 0. Consider to replace the expression with FDR below a certain threshold. (Comment #6)

Minor typos in P9L25 and P16L15. (Comment #7)

Author response to reviewer #2:

We thank the reviewer for their critical and thorough feedback of our manuscript which we believe have led to a much-improved manuscript, including new analyses on the enrichment of the enhancing high-grade glioma metabolome for plasma-derived metabolites. Please find below our responses to reviewer #2's comments and the changes made in the manuscript to address their comments.

Reviewer comment	Response	Manuscript
1. Overall, the analysis approaches are appropriate and the analyzed results	We thank the reviewer for their positive review of our analysis approaches and	Please see the end of this response for the new figures which have been added to the

Response to Reviewer Comments

support the main conclusions. One small caveat is that only one bloody CSF and one clean CSF are used to establish the blood metabolite set for GSEA analysis. It is understandable because these samples are very difficult to obtain. One suggestion is to use literature information to build a blood metabolite set.	results. Indeed, it is very difficult to obtain paired clean versus bloody CSF samples and our current data set only contained that one paired sample. We reviewed the literature to evaluate what data were available regarding differentially abundant metabolites between blood and CSF, particularly in regard to blood-contaminated CSF. Unfortunately, very little data existed that would enable building of a comprehensive blood metabolite set. Instead, we turned to our current CSF metabolomics data, which included global metabolomics for 28 unique intracranial CSF sample, collected at the time of surgery (including resection and shunt placement). We reasoned that we could take the cleanest and most bloody of these samples based on their heme content and evaluate their metabolic differences to create a pooled clean versus bloody CSF ranked list. As such, heme was utilized to rank the 26 CSF samples on a gradient from least bloody to most bloody, excluding the two paired CSF samples we had utilized in our original analyses. We then took the top (bloody, n=7) and bottom (clean) quartiles, ensuring that CSF samples were balanced in each category based on tumor grade, location, and other clinical variables. We then averaged the metabolites within each clean and bloody CSF and found the fold-change between bloody and clean CSFs. On visual inspection,	manuscript (Supplementary Figures S8 and S9). The following sentences have been added to the manuscript to describe these new analyses and results: “Twenty-six other intracranial CSF samples were collected with variable blood-contamination. Samples were centrifuged at 400 G for 10 minutes to remove red blood cells and other debris and aliquoted. For the paired bloody and clean CSF sample, the ranked metabolite list was based on metabolite fold-change (bloody/non-bloody) between samples, after filtering for metabolites present in both CSF and >90% of microdialysate samples. For the pooled bloody and clean CSF samples, CSF samples were ranked from greatest to least blood contamination based on heme content. Excluding the ependymoma sample, CSF from the top (bloody) and bottom (clean) quartiles were utilized (n=7 each), filtering for metabolites present >90% of microdialysate samples and >85% of CSF samples. Ranked metabolite lists were then generated based on the metabolite fold-change for the average of the bloody CSF samples versus the average of the clean CSF samples.” (Page 6, lines 9-19) “As it is technically challenging to obtain identically paired bloody versus clean CSF samples, we evaluated the
---	---	---

Response to Reviewer Comments

	this bloody-versus-clean metabolite list from the pooled CSF samples looked very similar to our n=1 paired bloody-versus-clean CSF list. GSEA confirmed that bloody metabolites in the pooled bloody-versus-clean CSF list were indeed enriched for bloody metabolites as defined by paired bloody-versus-clean CSF. Likewise, the clean CSF in the pooled bloody-versus-clean CSF ranked list was enriched for clean CSF in the paired CSF list, both at FDRs near 0. We then utilized the pooled bloody-versus-CSF ranked list to reproduce the analyses we had performed with the pooled bloody-versus-clean CSF. Using each patient's enhancing versus non-enhancing microdialysate, we again found that most enhancing microdialysates (6/7) were enriched for bloody CSF. This result was identical to our prior finding, although interestingly, the normalized enrichment scores were generally somewhat higher using pooled bloody-versus-clean CSF, suggesting more robust enrichment. Using the pooled bloody-versus-clean CSF, we found that 3/7 non-enhancing glioma microdialysates were enriched for clean CSF, as compared to 6/7 with the pooled bloody-versus-clean CSF analyses. As in Figure 5B, we recreated the ranked heatmap using the pooled bloody-versus-CSF ranked list and again found a higher abundance of bloody CSF-associated metabolites in enhancing tumor as	reproducibility of these results by performing the same analyses using pooled, rather than paired, bloody versus clean CSF samples (n=7 each; see methods). The pooled bloody versus clean CSF ranked list was positively enriched for bloody metabolites in the paired CSF sample and negatively enriched for clean metabolites ($FDR < 1 \times 10^{-5}$), suggesting robust metabolic similarities between both analysis methods (Supplementary Fig. S8A). Using enrichment analyses, we found that at least 6/7 patients had significant enrichment for at least one of (i) enhancing tumour in bloody CSF (6/7) or (ii) non-enhancing tumour in clean CSF (3/7) (Supplementary Fig. S8B). Overall, results for the enhancing tumor microdialysate were similar to those found using paired bloody-versus-clean CSF analyses, with more robust positive enrichment scores. Less robust enrichment results for clean CSF in non-enhancing tumor microdialysate using pooled, rather than paired, analyses may be due to heterogeneity in the clean CSF metabolome between non-paired patient samples. As in Fig. 5B, enhancing versus non-enhancing metabolites from Fig 4B demonstrated greater abundance of bloody CSF associated metabolites from the pooled analyses in enhancing tumour (Supplementary Fig. S8C). Enrichment analyses for enhancing tumour versus
--	--	---

Response to Reviewer Comments

	compared to non-enhancing tumor. Upon re-evaluating the enhancing-versus-brain and non-enhancing tumor-versus-brain microdialysates using our pooled CSF method, we also again found that there was significant enrichment for bloody CSF in the enhancing tumor microdialysate as compared to brain (9/9 samples). There was again modest enrichment for non-enhancing microdialysate when compared to brain (4/7, as with prior results). The enrichment of non-enhancing-versus-brain samples for bloody CSF was not always the same across the two different types of analyses, suggesting that samples may be enriched for different portions of the bloody CSF metabolome. In conclusion, the pooled bloody versus clean CSF analyses generally reproduced our original findings from the n=1 paired bloody versus clean CSF analyses, particularly regarding the enrichment of enhancing microdialysate for bloody CSF. We had originally hypothesized that results with pooled CSF analyses would likely be less robust than our paired analysis, as we have found that there is significant metabolic variation from patient-to-patient that requires an internal control. Nevertheless, we believe that the results from the pooled analyses are promising and support our original findings. We are continuing to obtain more paired bloody-versus-clean CSF analyses to evaluate the reproducibility of	brain again demonstrated enrichment for bloody CSF (9/9 patients; Supplementary Fig. S9A), while non-enhancing tumour versus brain microdialysates had variable enrichment for bloody CSF (4/7 patients (Supplementary Fig. S9B), similar to the paired bloody-versus-clean CSF analyses.” (Pages 13-14, Lines 18-34, 1-2)
--	---	---

Response to Reviewer Comments

	our findings in a larger cohort of patients, although it is very technically difficult to obtain bloody-versus-clean CSF from an identical location in a patient. As such, we have added in the new analyses and results to the manuscript as described in the next column, which we believe further strengthens the manuscript's conclusion.	
2. "The discovery of GAA being a differential metabolite is an important finding for the manuscript. The authors mention GAMT deficiency could result to GAA accumulation. It would be great to include patient data for GAMT deficiency, or to discuss GAMT's role in the current finding."	We agree with the reviewer that the discovery of GAA is an important finding of our manuscript. GAA can indeed accumulate in patients with GAMT deficiency; however, creatine production is diminished in these patients. We have not found any differences in creatine levels between enhancing tumor and brain where there was the most profound difference in GAA levels. As such, we do not believe that GAMT deficiency explains the elevated GAA seen within tumor as compared to brain. Instead, based on pathway analyses, we realized that ornithine was co-produced with GAA via AGAT. Based on the well-established role of ornithine and its derivatives, polyamines, in cancer biology, we hypothesize that GAA is a byproduct of elevated polyamine synthesis. We describe in the manuscript how we will be testing this hypothesis in a clinical trial of DFMO+AMXT.	Relevant portions of manuscript: "GAA is a precursor for creatine, levels of which were slightly elevated in non-enhancing (2.25x; p=0.03), but not enhancing tumour (Table 2). Interestingly, GAA is co-produced with ornithine, which was slightly higher in enhancing tumour when compared to non-enhancing tumour (2.36x, p=0.03) and brain (3.07x, p=0.004) (Table 2)... To determine if GAA can be used as a dynamic biomarker of ODC production in situ, we are opening a post-operative microdialysis study under an IND, monitoring microdialysate levels of GAA in unresectable regions of residual tumour with or without DFMO +/- AMXT1501." (Page 17, lines 11-28)
3. "P3L18: Cited ref 13-15 are more than ten years from now. Citing more recent publications on the related topic may further highlight the technical and biological	We thank the reviewer for this suggestion. While we agree that it would be of interest to cite more recent publications, we have been unable to find more recent studies utilized	N/A

Response to Reviewer Comments

advancement in the current manuscript.”	microdialysis intra-operatively for the purpose of collecting metabolites during standard-of-care resections. There are a number of new studies in the post-operative setting, most often for pharmacokinetic studies. As such, we have not added any more recent citations to the paper. This lack of new data in the past ten years highlights the technical and biological advancements regarding glioma metabolism afforded by our intra-operative high-molecular weight microdialysis study.	
4. “P7L23-P8L12. Some of the sentences may go to the introduction or method section so that the results could stand out better.”	We thank the reviewer for this helpful suggestion and agree that this section should be moved to focus on results within the section. We have moved portions of that excerpt to the introduction, as seen in the next column.	Revision: “Microdialysis has been used to quantify human extracellular biomarkers of traumatic and hypoxic brain injury in neurocritical care units. This is most commonly performed in the post-operative setting with low-molecular weight catheters, including an FDA-approved system consisting of 20 kDA catheters perfused at 0.3 μL/min using the M-dialysis 106 pump. Microdialysis is also a well-established method to quantify central nervous system (CNS) drug delivery in early phase clinical trials⁸⁻¹⁰ and has been used longitudinally to study metabolites present in human gliomas when compared to adjacent brain^{11,12}. While a limited subset of patients may be willing to undergo post-operative microdialysis, it is usually hoped that there will be minimal tumour left to sample following surgery, often preventing sampling from multiple regions of the glioma. As such, intra-operative microdialysis may

Response to Reviewer Comments

		be of interest for evaluating diverse regions of gliomas.” (Page 4, line 13-23)
5. “P9L26. Is elevated 2HG in blood a biomarker for more invasive IDH mutant gliomas? If literatures support this idea, it can support that 2HG level high in non-enhancing region than enhancing region is due to the 2HG diffusion to the disrupted blood brain barrier.”	The reviewer poses a very interesting question relevant to many of our ongoing studies. While elevated 2-hydroxyglutarate has been found in some plasma and urine studies, to our knowledge, there is no data correlating the abundance of 2-HG with the invasiveness/grade of the IDH-mutant gliomas. We agree that it would be highly relevant to determine whether elevated peripheral 2-hydroxyglutarate correlates with the relative level of blood-brain barrier disruption. Tangentially related, we have an ongoing project evaluating the utility of CSF 2-hydroxyglutarate as a diagnostic and monitoring biomarker for IDH-mutant glioma; along with CSF samples, we are now routinely obtaining paired plasma samples for each patient. As such, we hope that we will be able to answer this question in a separate manuscript focusing specifically on the role of 2-HG as an IDH-mutant glioma biomarker and any correlations thereof to aggressivity/blood-brain barrier disruption that would support the reviewer’s and our hypothesis.	No changes.
6. P13L5 & L16. FDR = 0.000 or FDRs = 0 may be the results of software omitting digits after a few zeros. FDR could be very small, such as 10⁻⁶, but should not be 0. Consider to replace the expression with FDR below a certain threshold.	We thank the reviewer for this suggestion. The outputs of FDR=0 were indeed obtained directly from the software where they are written as such. Based on the other FDRs, we have replaced the FDR=0 with a threshold of FDR<1x10⁻⁵, given that	Any FDR written as 0.E+00 in Supplemental Tables 2-4 has been replaced with <1E-05. Addition: “Graphs were generated using GraphPad PRISM 9.1. FDR_≤0.05 was considered statistically significant for enrichment

Response to Reviewer Comments

	other FDRs, even when highly significant, were above this value.	analysis. FDR values given as 0 by GSEA were reported as $<1 \times 10^{-5}$." (Page 7, lines 16-18) Edit: "Leveraging these individual patient-level data, these analyses proved highly robust across patients with a False Discovery Rate (FDR) of $<1 \times 10^{-5}$ for most interpatient comparisons of enhancing tumour versus brain metabolites." (Page 11, lines 14-16) Edit: "...could be consistently found across most patients (most FDRs $<1 \times 10^{-5}$, except GBM^{WT1}, Fig. 3Bii; Supplementary Table 3)" (Page 11, lines 26-27)
7. "Minor typos in P9L25 and P16L15."	We thank the reviewer for their thorough reading of the manuscript and apologize for these errors, which have now been corrected.	Correction: "This observation raised the possibility that 2-HG could be lost down its concentration gradient into systemic circulation." (Page 8, lines 32-34) Correction: "For example, although glucose was identified as utilised by "glioma" in their data, glucose readily crosses the BBB and is highly utilised by brain, making fluorodeoxyglucose (FDG)-positron emission tomography (PET) unhelpful to discriminate tumour from brain." (Page 14, 17-21)

New Figures:

Response to Reviewer Comments

Supplementary Figure S8. Enrichment of paired enhancing versus non-enhancing tumour microdialysates for pooled bloody versus clean CSF samples.

(A) Enrichment plots for bloody metabolites from paired bloody versus clean CSF in pooled blood versus clean CSF samples ($n=6$ each). (B) Enrichment plots for clean metabolites from paired bloody versus clean CSF in pooled blood versus clean CSF samples ($n=7$ each). (C) Enrichment analysis was utilised to determine the enrichment of pooled bloody versus clean CSF for patient's enhancing vs. non-enhancing tumour catheters. Positive normalised enrichment scores (NES) indicate metabolic similarities to high-plasma (bloody) CSF; negative NES indicate metabolic

Response to Reviewer Comments

similarities to low-plasma (clean) CSF (* = $FDR \leq 0.05$). (D) The rank of each metabolite from the significant enhancing versus non-enhancing glioma metabolite in Figure 4B is shown for each patient (E: purple, rank 1, to NE: orange, rank 162), along with the metabolite's rank in the pooled bloody vs. clean CSF ranked fold-change list (bloody: red, 1 – clean: blue, 153).

Supplementary Figure S9. Plasma-derived metabolites in enhancing and non-enhancing versus brain microdialysates based on pooled bloody vs. clean CSF samples.

Enrichment Analysis was utilised to determine the enrichment of each patient's (A) enhancing tumour or (B) non-enhancing tumour versus brain microdialysates for bloody versus clean CSF based on a ranked list of pooled bloody versus clean CSF samples (n=7 each). Positive normalised enrichment scores (NES) indicate metabolic similarities to bloody CSF; negative enrichment scores indicate metabolic similarities to clean CSF (* = $FDR < 0.05$).

Reviewers' comments:

Reviewer #1 (Remarks to the Author):

The study aimed to analyze the extracellular metabolome of glioma tumors in patients, focusing on the influence of blood-brain barrier disruption on the presence of metabolites. The results revealed that the metabolites detected were predominantly influenced by the disruption of the blood-brain barrier, with enhanced tumor regions showing a significant enrichment of plasma-associated metabolites. In order to collect extracellular analytes, such as metabolites and specific drugs, from both enhancing and non-enhancing tumor regions, intraoperative microdialysis was employed. Through this approach, the researchers were able to identify a consistent metabolome in enhancing gliomas compared to non-enhancing gliomas, and they observed similar patterns within individual patients. These findings suggest that investigations into the metabolic microenvironment of human gliomas in situ should take into account the disruption of the blood-brain barrier as a crucial factor. Therefore, the study provides valuable insights into the relationship between blood-brain barrier disruption and glioma aggressiveness, highlighting the potential of intraoperative microdialysis as a tool for studying the extracellular metabolome of live human gliomas.

However, it is important to acknowledge that this study is based on a limited patient cohort, and there is a scarcity of information regarding the metabolic microenvironment of human gliomas in situ. To further establish the reliability of the method used in this study, it would be beneficial for the authors to compare the metabolite profiles with other published datasets. Additionally, the dataset lacks information concerning the impact of IDH mutations on the metabolome of gliomas. Therefore, it would be worthwhile for the authors to investigate the relative abundance of plasma-derived metabolites in both contrast-enhancing and non-enhancing gliomas, in order to determine if these metabolites contribute to the extracellular metabolome of enhancing gliomas. Another aspect that requires attention is the clinical relevance of this study. Although it is intriguing to observe the metabolic differences between non-enhancing and enhancing tumors in the brain, the fact that enhancing tumors acquire more plasma-associated metabolites is not surprising. Thus, the study should emphasize and strengthen the therapeutic and clinical value of these findings.

Reviewer #3 (Remarks to the Author):

Overall, this reviewer thinks the claim "Blood-Brain Barrier Disruption Define the Glioma Extracellular Metabolome" is mainly supported by the enrichment analysis based on bloody metabolite set from bloody-CSF vs clean-CSF. The additional analyses and supplementary figures help clarify previous comments and strengthen the claim.

One follow-up question regarding to the GAA and the 2HG comment response: the authors show 2HG is higher in non-enhancing tumor than in enhancing tumor; while the newly discovered differential metabolite GAA is higher in enhancing tumor than in non-enhancing tumor. The 2HG difference can be explained by leakage to the blood stream via disrupted BBB in enhancing tumor, which supports the main claim of the manuscript. If both 2HG and GAA are produced by tumors, the reviewer is curious to know any rationale that may explain GAA is higher in enhancing tumor. The integration of these three aspects will further strengthen the manuscript.

Response to Reviewer Comments - COMMSBIO-23-0765A

Reviewer #1 (Remarks to the Author):

The study aimed to analyze the extracellular metabolome of glioma tumors in patients, focusing on the influence of blood-brain barrier disruption on the presence of metabolites. The results revealed that the metabolites detected were predominantly influenced by the disruption of the blood-brain barrier, with enhanced tumor regions showing a significant enrichment of plasma-associated metabolites. In order to collect extracellular analytes, such as metabolites and specific drugs, from both enhancing and non-enhancing tumor regions, intraoperative microdialysis was employed. Through this approach, the researchers were able to identify a consistent metabolome in enhancing gliomas compared to non-enhancing gliomas, and they observed similar patterns within individual patients. These findings suggest that investigations into the metabolic microenvironment of human gliomas *in situ* should take into account the disruption of the blood-brain barrier as a crucial factor. Therefore, the study provides valuable insights into the relationship between blood-brain barrier disruption and glioma aggressiveness, highlighting the potential of intraoperative microdialysis as a tool for studying the extracellular metabolome of live human gliomas.

However, it is important to acknowledge that this study is based on a limited patient cohort, and there is a scarcity of information regarding the metabolic microenvironment of human gliomas *in situ*. To further establish the reliability of the method used in this study, it would be beneficial for the authors to compare the metabolite profiles with other published datasets. Additionally, the dataset lacks information concerning the impact of IDH mutations on the metabolome of gliomas. Therefore, it would be worthwhile for the authors to investigate the relative abundance of plasma-derived metabolites in both contrast-enhancing and non-enhancing gliomas, in order to determine if these metabolites contribute to the extracellular metabolome of enhancing gliomas. Another aspect that requires attention is the clinical relevance of this study. Although it is intriguing to observe the metabolic differences between non-enhancing and enhancing tumors in the brain, the fact that enhancing tumors acquire more plasma-associated metabolites is not surprising. Thus, the study should emphasize and strengthen the therapeutic and clinical value of these findings.

Author response to reviewer #1 comments:

We thank the reviewer for their positive review of our revised manuscript, which they state “provides valuable insights into the relationship between blood-brain barrier disruption and glioma aggressiveness.” We also thank the reviewer for their helpful commentary on points of the manuscript that should be emphasized or strengthened. Please see below for our responses to the reviewer’s comments.

Reviewer comment	Author response	Manuscript
“It is important to acknowledge that this study is based on a limited patient cohort , and there is a scarcity of information regarding the metabolic microenvironment of human gliomas in situ .”	We fully agree with the reviewer that the study is based on a limited patient cohort and that there is limited information on the metabolic microenvironment of human gliomas in situ . We have described these points in the portions of the manuscript highlighted in the right column.	Regarding the limited patient cohort: “We acknowledge the limitations of this study, including the small sample size of fourteen patients across fifteen surgeries, which may limit the generalizability of our results despite robust metabolic similarities across two heterogeneous cohorts of patients.” (Page 18; lines 4-6)

		Addition: “Although glioma metabolism is increasingly scrutinised for therapeutic targets, few strategies currently exist to interrogate the metabolic microenvironment of human gliomas in situ, and there is a relative paucity of global metabolomic data from live human gliomas.” (Page 4, lines 11-13)
“It would be beneficial for the authors to compare the metabolite profiles with other published datasets.”	We fully agree with the reviewer that this is an important consideration given the relative novelty of this method for intraoperative sampling of gliomas. Few datasets exist that have queried gliomas in the same manner that we have, particularly in regard to global metabolomic analyses rather than evaluation of a distinct small subset of metabolites. However, we did highlight and comment on one study that also evaluated a diverse array of metabolites via microdialysis in glioblastoma, albeit in the post-operative setting. Their findings were in line with our findings, suggesting the reproducibility of microdialysis for evaluation of the global metabolome. Of note, we also highlighted how our study contributes to the literature rather than just reinforcing data found in other prior studies. We have also added references here to intra-operative microdialysis studies that performed metabolomics on a much smaller subset of metabolites, although these more limited analyses still confirmed much of our findings.	“Our rank-based analyses revealed an enhancing glioma extracellular metabolome that was consistent across primary and recurrent IDH-WT and mutant lesions, as well as an H3K27M-mutant tumour (Fig. 3A(i-ii)). This signature appeared comparable to that previously obtained in the context of intra-or-post-operative microdialysis¹³⁻¹⁷, consistent with reproducibility of the enhancing glioma metabolome across clinical and technical variables (Supplementary Table 1), research teams and contexts spanning intra-operative versus post-operative sampling. However, our data demonstrate for the first time that most of the metabolites in the enhancing glioma extracellular metabolome are absent from the non-enhancing glioma metabolome.” (Page 15-16; lines 27-34, 1)
“Additionally, the dataset lacks information concerning	We agree with the reviewer that it is very important to	Discussion on IDH-mutant versus wild-type metabolomic

the impact of IDH mutations on the metabolome of gliomas.”	consider the impact of IDH mutations on the metabolome of gliomas, particularly given that our patient cohort contains a mix of patients with IDH-mutant or wild-type gliomas. While we confirmed the presence of elevated 2-hydroxyglutarate in the microdialysate of IDH-mutant tumors, but not IDH-wild type tumors, as compared to brain, we did not see a difference in the global metabolome of IDH-mutant versus wild-type tumors based on our Spearman correlation analyses. These results can be found in Figure 1C-D and Figure 2A (described on pages 7-9) of our manuscript and we discuss the meaning of these results in the highlighted section of the manuscript in the next column.	results can be found in full on page 15, lines 5-23. Of note: “We observed elevated 2-HG in the microdialysate of IDH-mutant tumours via both targeted (D vs L-2-HG) and untargeted (total D+L-2HG) analysis (Fig. 1C; Supplementary Fig. 2). ... Interestingly, 2-HG was 5-12x lower in the most avidly enhancing tumour region (n=3) (Fig. 1D), suggesting that some tumour metabolites may be lost down a disrupted blood brain barrier—a consideration of potential importance for longitudinal pharmacodynamic monitoring of tumour interstitial fluid. Importantly, although IDH-mutations are thought to drive early tumourigenic metabolic and epigenetic changes^{32,38}, the global extracellular metabolomes of enhancing IDH-mutant versus IDH-WT grade 4 astrocytomas were otherwise indistinguishable (Fig. 2A, 3A). Recently published work from over 200 glioma tissues demonstrated less separation between IDH-mutant grade 4 astrocytomas from GBMs²⁷, than lower grade IDH-mutant gliomas, potentially suggesting metabolic convergence of high-grade gliomas. Greatest similarities were seen across patients in the extracellular metabolome of contrast-enhancing tumours, which could be expected if improved availability of plasma-associated metabolites impacts tumour metabolism.”
“It would be worthwhile for the authors to investigate the relative abundance of plasma-derived metabolites in	We thank the reviewer for this helpful suggestion which we also think is of great importance to this manuscript.	Excerpt: “In the 7 patients for whom paired enhancing and non-enhancing microdialysates were

both contrast-enhancing and non-enhancing gliomas, in order to determine if these metabolites contribute to the extracellular metabolome of enhancing gliomas.”	In line with the reviewer's comment, since high-grade gliomas often contain both enhancing and non-enhancing portions, we sampled the extracellular metabolome of both the enhancing and non-enhancing portions of a patient's glioma as we believe it is important to have internal controls for each patient. Through these analyses of the enhancing versus non-enhancing metabolome in intra-patient paired samples, we found that the enhancing glioma metabolome, but not the non-enhancing glioma metabolome of that same patient, was enriched for plasma-derived metabolites (Figure 5; Supplemental figure S8). This enrichment varied from patient to patient, with some patients having an enhancing glioma metabolome that was far more enriched for plasma-derived metabolites than others (for example, patient Astro^{4-mut3} versus GBM^{WT2}). Of note, we also performed analyses of the enhancing glioma versus brain and non-enhancing glioma versus brain metabolomes to find their relative enrichment for plasma-derived metabolites. We expected that there could be variable enrichment for plasma-derived metabolites in the non-enhancing portions of these tumors when they were compared to brain based on the potential for sub-radiographic BBB disruption (preventing enhancement with a gadolinium contrast-agent), or perhaps due to	obtained, we first compared the relative enrichment of enhancing versus non-enhancing brain catheters for bloody versus clean CSF using each patient's ranked enhancing versus non-enhancing list (Cath. X/Y). Enrichment analyses demonstrated significant enrichment in 7/7 patients for at least one of (i) enhancing tumour enriched in bloody CSF, or (ii) non-enhancing tumour enriched in clean CSF; 5/7 patients showed significant enrichment in both analyses (FDR<0.05) (Fig. 5A). Evaluation of specific enhancing versus non-enhancing tumour-associated metabolites from Fig 4B revealed higher abundance of bloody CSF-associated metabolites in enhancing tumour (Fig. 5B). As expected, performing enrichment analyses on enhancing glioma versus brain ranked list again demonstrated that enhancing glioma catheters were enriched for bloody CSF in 9/9 patients (Supplementary Fig. S7). Results in non-enhancing tumour suggested a modest enrichment of bloody CSF-associated metabolites in some catheters, perhaps suggestive of either some sub-radiographic BBB disruption²⁹, or metabolite diffusion from adjacent enhancing regions.” (Page 13; 4-17) (We also reproduced these results using pooled bloody versus clean CSF analyses and found similar results; these are described on page
--	--	--

	metabolite diffusion from nearby enhancing areas. The question of enhancing and non-enhancing glioma can be addressed in 2 ways: internally controlled samples within a single patient that contains both non-enhancing and enhancing tissue, and separate comparison of enhancing tumors versus non-enhancing tumors. We found that each patient's microdialysate metabolome tended to be quite different from each other patient. As such, it was necessary to have an internal control. Through analyses of the enhancing versus non-enhancing metabolome in intra-patient paired samples, we found that the enhancing glioma metabolome, but not the non-enhancing glioma metabolome of that same patient, was enriched for plasma-derived metabolites (Figure 5; Supplemental figure S8). This degree of enrichment varied from patient to patient, with some patients having an enhancing glioma metabolome that was far more enriched for plasma-derived metabolites than others (for example, patient Astro^{4-mut3} versus GBM^{WT2}). Of note, we also performed analyses of the enhancing glioma versus brain and non-enhancing glioma versus brain metabolomes to identify their relative enrichment for plasma-derived metabolites. We expected that there could be variable enrichment for plasma-derived metabolites in the non-enhancing portions of	13-14, lines 18-34 and 1-4 of the manuscript).
--	--	---

	these tumors when they were compared to brain based on the potential for sub-radiographic BBB disruption (preventing enhancement with a gadolinium contrast-agent), or perhaps due to metabolite diffusion from nearby enhancing areas. Based on the wording of the reviewers' comment ("...and non-enhancing gliomas") we appreciate that while enhancing gliomas are evaluated in detail, the reviewer is specifically ALSO interested in non-enhancing gliomas, for which we did not originally provide detailed analysis, given lack of an enhancing internal control for comparison. To address this, we have now performed new analyses of the tumors with only non-enhancing tumor catheters versus adjacent brain (five patients). These results are provided below this table for the reviewer's reference. Enrichment analyses revealed highly variable enrichment of the non-enhancing tumor catheters for plasma-derived metabolites, with the strongest and most consistent enrichment (2/2 non-enhancing catheters per patient) noted in patients Oligo³¹ and GemAstro^{3-mut}. It should be noted that these results were highly variable for tumors wherein only non-enhancing disease was sampled (mainly non-enhancing tumors). In some patients, the brain catheter appeared relatively more enriched for plasma-derived metabolites than the FLAIR	
--	---	--

	catheter, e.g. one of the two brain versus non-enhancing tumor comparisons in patients Oligo³³ and MolecGBM^{WT}, and both tumor vs brain comparisons for Oligo³². As such, we must conclude that non-enhancing gliomas are NOT consistently enriched for plasma, which is different than for enhancing tumors, where results were extremely consistent. The variable positive and negative enrichment, however, also raises several questions and possible hypotheses, none of which we have sufficient data to prove or disprove, but are provided here for transparency in response to the reviewer's query: (1) Catheter insertion could induce a variable amount of blood-brain barrier disruption. If true, this appears not to have impacted our paired analysis of enhancing versus non-enhancing tumor or enhancing versus brain comparisons within individual patients—likely since these signatures were sufficiently robust that they were not impacted by a small amount of catheter-related BBB disruption. However, when evaluating a very weak signature, such as brain versus non-enhancing tumor, the previously imperceptible impact of inconsistent catheter placement-associated BBB disruption may stochastically impact results. (2) Of note, the specific patients with non-enhancing tumors included in this manuscript generally had very diffuse disease, and therefore	
--	---	--

	contained some amount of FLAIR signal even in the relatively “Brain” catheter. It is therefore not inconceivable that some early amount of tumor-associated BBB disruption could have been present in some regions of some tumors, given known tumor heterogeneity—even if those were not the areas regarded as the most “FLAIR-positive” regions. (3) It is relevant to provide a specific clarification regarding GemAstro^{3-mut}: while this patient had an enhancing portion to their tumor, we were not able to sample it due to its highly fibrous nature preventing penetration of the tumor by the microdialysis catheter. As such, we sampled two non-enhancing portions of the tumor. It is possible that the enrichment for plasma-derived metabolites in the non-enhancing areas of this tumor could be due to diffusion of plasma-associated metabolites from the nearby enhancing areas that could not be sampled into the FLAIR regions. This would not explain the similar results seen in patient Oligo³¹, who had a completely non-enhancing oligodendroglioma. But we have noted that this patient’s tumor generally appeared more analogous to results from astrocytomas. The reason for this remains unclear, and we are currently focusing on performing further studies of patients with low-grade gliomas to better understand the heterogeneity	
--	--	--

	observed across such lesions. Due to the speculative and extremely preliminary nature of these analyses and putative explanations, we did not consider it appropriate to include these analyses within the manuscript, which focuses more on the tumors for which we have an internal control of enhancing and non-enhancing regions. We hope to publish a separate manuscript on non-enhancing gliomas once we have a large enough sample size with enough data to more definitively answer the possible rationales, and will include these analyses at that time. We thank the reviewer for this very important question which we hope will lead to novel metabolic insights on non-enhancing gliomas.	
“Another aspect that requires attention is the clinical relevance of this study. Although it is intriguing to observe the metabolic differences between non-enhancing and enhancing tumors in the brain, the fact that enhancing tumors acquire more plasma-associated metabolites is not surprising. Thus, the study should emphasize and strengthen the therapeutic and clinical value of these findings.”	We fully agree with the reviewer that our findings are intuitive based on the aggressivity of these gliomas. Based on this, we also agree that it is important to consider how these findings could be therapeutically targeted and their clinical relevance. In regard to the clinical relevance, we emphasize in our discussion how these metabolic findings align with how radiographic contrast-enhancement is a well-established clinical parameters of glioma aggressivity and worse prognosis (excerpt in the next column). We also emphasize mechanisms for therapeutic targeting of this pathway, namely the need for more	Excerpt: “Given the established importance of multiple identified plasma- and enhancing tumour-associated metabolites, including amino acids and carnitines to cancer biology⁴⁸⁻⁵⁰, BBB disruption itself may underlie the metabolic similarities across molecularly diverse HGGs by providing a continuous source of protumourigenic metabolites. Anecdotally, our patient with a completely non-enhancing molecular GBM had evidence of this lesion on a CT obtained 5 years prior to diagnosis—consistent with a lower rate of growth than patients with enhancing lesions. Indeed, the adverse prognostic impact of a

	effective blood-brain barrier sealing agents that could limit access to these pro-tumorigenic plasma-derived metabolites (excerpt in the next column). We also discuss a specific pathway of interest based on our finding of guanidinoacetate, which is the polyamine pathway for which we have initiated a phase 0 study with polyamine-targeting therapies, DFMO+AMXT-1501.	contrast-enhancement in gliomas, including histologically low-grade tumours, is well recognised^{51,52}, as is the positive impact on survival of gross total resection of enhancing tumour across all molecularly defined GBM subgroups and ages⁵³. As such, our findings suggest that blood brain barrier disruption itself may transform the extracellular glioma metabolome into a wellspring of nutrients for accelerated growth, particularly in enhancing tumour regions.” (Page 16, lines 12-23) Excerpt: “It is possible that the induction of blood-brain barrier disruption by proliferating gliomas⁵⁸ may represent an adaptive response to nutrient deprivation observed in gliomas⁵⁹⁻⁶¹, catalysing a feedforward cycle of nutrient utilisation and recruitment. Mechanistic studies selectively manipulating BBB permeability or extracellular plasma abundance will be required to dissect these interactions. If our hypothesis of disrupted BBB-mediated tumour aggressiveness is correct, maintaining and restoring BBB integrity may prove an independently important therapeutic goal, distinct from angiogenesis-blocking agents such as bevacizumab⁶².” (Page 16-17, lines 30-34 and 1-2) Excerpt: “If true and based on the substantial fold-change,
--	---	---

		extracellular GAA could serve as a biomarker of not only local glioma abundance but also pharmacodynamic response to ODC blockade. ODC blockade with Difluoromethylornithine (DFMO) remains in clinical trials for glioma (NCT02796261); though upregulation of polyamine transporters may mediate metabolic compensation leading to improved survival with dual blockade of ODC and polyamine transporters (via AMXT-1501), in preclinical models of diffuse midline gliomas and other cancers^{66,67}. To determine if GAA can be used as a dynamic biomarker of ODC production in situ, we are opening a post-operative microdialysis study under an IND, monitoring microdialysate levels of GAA in unresectable regions of residual tumour with or without DFMO +/- AMXT1501." (Page 17; lines 23-33)
--	--	---

NEW ANALYSES for NON-ENHANCING VERSUS BRAIN GLIOMAS

Plasma-derived metabolites in completely non-enhancing tumors versus brain microdialysates based on paired bloody versus clean CSF samples. For the five completely non-enhancing tumors, enrichment analysis was utilised to determine the enrichment of each patient's two non-enhancing tumour catheters (Cath X: NE^X or Cath Y: NE^Y) versus brain microdialysates for bloody versus clean CSF based on a ranked list of paired bloody versus clean CSF samples (n=7 each). Positive normalised enrichment scores (NES) indicate metabolic similarities to bloody CSF; negative enrichment scores indicate metabolic similarities to clean CSF (*= FDR<0.05).

Plasma-derived metabolites in completely non-enhancing tumors versus brain microdialysates based on pooled bloody versus clean CSF samples. For the five completely non-enhancing tumors, enrichment analysis was utilised to determine the enrichment of each patient's two non-enhancing tumour catheters (Cath X: NE^X or Cath Y: NE^Y) versus brain microdialysates for bloody versus clean CSF based on a ranked list of pooled bloody versus clean CSF samples (n=7 each). Positive normalised enrichment scores (NES) indicate metabolic similarities to bloody CSF; negative enrichment scores indicate metabolic similarities to clean CSF (*= FDR<0.05).

Response to Reviewer Comments - COMMSBIO-23-0765A

Reviewer #3 (Remarks to the Author):

Overall, this reviewer thinks the claim “Blood-Brain Barrier Disruption Define the Glioma Extracellular Metabolome” is mainly supported by the enrichment analysis based on bloody metabolite set from bloody-CSF vs clean-CSF. The additional analyses and supplementary figures help clarify previous comments and strengthen the claim.

One follow-up question regarding to the GAA and the 2HG comment response: the authors show 2HG is higher in non-enhancing tumor than in enhancing tumor; while the newly discovered differential metabolite GAA is higher in enhancing tumor than in non-enhancing tumor. The 2HG difference can be explained by leakage to the blood stream via disrupted BBB in enhancing tumor, which supports the main claim of the manuscript. If both 2HG and GAA are produced by tumors, the reviewer is curious to know any rationale that may explain GAA is higher in enhancing tumor. The integration of these three aspects will further strengthen the manuscript.

Author response to reviewer #2 comments:

We thank the reviewer for their positive comments on the revisions made to our manuscript which have helped “clarify previous comments and strengthen the claims.” The reviewer astutely points to an interesting portion of our data, namely that while 2-HG is higher in non-enhancing tumor than enhancing tumor, GAA is more elevated in enhancing than non-enhancing tumor. While we hypothesized that 2-HG could leak out of disrupted portion of the tumor, the same statement cannot be made for GAA. We hypothesize that this discrepancy could be due to a far greater tumor-versus-blood gradient for 2-hydroxyglutarate than for GAA. As described in the manuscript, we speculate that a certain amount of the GAA in enhancing tumor, as compared to non-enhancing tumor, is due to blood-brain barrier disruption; however, the average fold-change between enhancing tumor vs. brain and non-enhancing tumor versus brain is disproportionately high, suggesting potential production by the tumor (excerpt below). Since 2-hydroxyglutarate is uniquely overproduced by IDH-mutant tumors, there is a significant gradient between the BBB-disrupted tumor and brain, allowing for rapid and significant dissipation from the tumor microenvironment. In contrast, we found that there was a low abundance of GAA in bloody CSF as compared to it being undetectable in clean CSF, suggesting that a small portion of GAA abundance in the enhancing microdialysate could be plasma-derived. As such, there is likely less of a tumor versus brain gradient for GAA, diminishing its rate of diffusion from the tumor, hence why it remains more elevated in enhancing tumor. We have added this rationale to the discussion section of our manuscript per below (addition below).

Excerpt/addition: “Although GAA was present at a low abundance in bloody CSF and not detected in clean CSF, its disproportionately high fold change in both enhancing and non-enhancing tumour versus brain (126.32x, and 13.63x, respectively; **Figure 4A, D(i), Table 2**) suggest it may be produced by the tumour. **Of note, while some GAA in enhancing tumor may be plasma-derived, 2-HG is uniquely aberrantly created by IDH-mutant tumors. There would thus be a less profound tumor-versus-plasma concentration gradient for GAA as compared to 2-HG, perhaps explaining why 2-HG, but not GAA, was more elevated in non-enhancing than enhancing tumour microdialysate.**” (Page 17; lines 5-12)